# MEMORIZATION IN SELF-SUPERVISED LEARNING IMPROVES DOWNSTREAM GENERALIZATION

**Wenhao Wang**[*1]**, Muhammad Ahmad Kaleem**[*2]**, Adam Dziedzic**[*†1]**,**
**Michael Backes**[1]**, Nicolas Papernot**[2]**, Franziska Boenisch**[†1]
[1]CISPA, [2]University of Toronto and Vector Institute

## ABSTRACT

Self-supervised learning (SSL) has recently received significant attention due to its ability to train high-performance encoders purely on unlabeled data—often scraped from the internet. This data can still be sensitive and empirical evidence suggests that SSL encoders memorize private information of their training data and can disclose them at inference time. Since existing theoretical definitions of memorization from supervised learning rely on labels, they do not transfer to SSL. To address this gap, we propose `SSLMem`, a framework for defining memorization within SSL. Our definition compares the difference in alignment of representations for data points and their augmented views returned by both encoders that were trained on these data points and encoders that were not. Through comprehensive empirical analysis on diverse encoder architectures and datasets we highlight that even though SSL relies on large datasets and strong augmentations—both known in supervised learning as regularization techniques that reduce overfitting—still significant fractions of training data points experience high memorization. Through our empirical results, we show that this memorization is essential for encoders to achieve higher generalization performance on different downstream tasks.

## 1 INTRODUCTION

In recent years, self-supervised learning (SSL) has emerged as a new potent learning paradigm. SSL encoders can be trained without reliance on labeled data, which is often hard and expensive to obtain. Instead, SSL leverages the existence of large amounts of unlabeled data—often scraped from the internet—to obtain state-of-the-art performance in various domains, ranging from computer vision (He et al., 2022; Chen et al., 2020; Chen & He, 2021; Caron et al., 2021) to natural language processing (Devlin et al., 2018; Radford et al.).

Empirical studies suggest that SSL encoders can disclose information about their training data at inference time (Meehan et al., 2023). An unintended revelation of private information is often associated to machine learning models' ability to memorize their training data (Zhang et al., 2016; Arpit et al., 2017; Chatterjee, 2018; Carlini et al., 2019; 2021; 2022). Studies in supervised learning revealed that mainly mislabeled samples, outliers (Bartlett et al., 2020; Feldman, 2020; Feldman & Zhang, 2020), or data points that were seen towards the end of training (Jagielski et al., 2022) are memorized and why memorization is crucial for the success of learning (Feldman, 2020; Feldman & Zhang, 2020; Arpit et al., 2017; Tirumala et al., 2022). Additionally, it was found that in supervised learning memorization happens in the feature extractor (encoder) layers (Feldman & Zhang, 2020; Maini et al., 2023). Those are exactly the type of layers that SSL trains. Yet, given that SSL differs significantly from supervised learning in terms of learning objective, data processing, and augmentation strength, it remains unclear whether the trends from supervised learning transfer to the self-supervised learning.

To date, a key limitation for studying memorization in SSL lies in the fact that the theoretical definitions from supervised learning (Feldman, 2020) cannot be applied since they rely on class labels which are not available in SSL. Existing empirical approaches to assess privacy leakage in

---

[*]Equal contribution. Correspondence to: adam.dziedzic@sprintml.com and boenisch@cispa.de.
[†]Part of the work was done while the authors were at the University of Toronto and the Vector Institute.

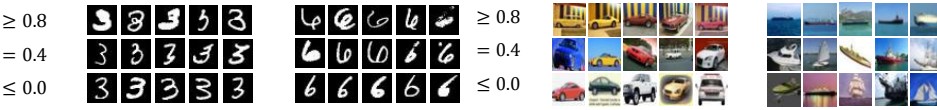

|  | (a) MNIST: class *3* and *6*. | (b) CIFAR10: class *automobile* and *ship*. |

Figure 1: **Examples of data with different levels of memorization.** Higher memorization scores indicate stronger memorization. We observe that outliers and atypical examples experience higher memorization than more standard samples. Results are obtained on a ViT-tiny, trained with MAE.

SSL are equally unsuited to study general SSL memorization since they still assume the existence of labels (Meehan et al., 2023). Membership inference attacks (Shokri et al., 2017) are highly related to memorization. Yet, to date, membership inference in SSL has been solely used to study privacy risks by answering the question whether or not a particular training data point was used to train a given encoder (Liu et al., 2021). Given that memorization is not a binary concept, nor a property of a particular trained encoder, our work goes beyond prior considerations and uses memorization to study general properties and behavior of SSL methods, such as their generalization capabilities.

Deriving a definition of memorization tailored to SSL and *general over all methods* comes with a severe challenge. In supervised learning, all methods directly optimize for the same objective (high confidence prediction on correct class labels) which creates a strong direct signal that can be measured to assess memorization (Feldman, 2020). In contrast, different SSL methods solve different optimization tasks in their respective projection spaces. Some methods, for example, minimize the reconstruction loss on an added decoder (He et al., 2022), others minimize a contrastive or non-contrastive loss in an additional projection space (Chen et al., 2020; Chen & He, 2021). None of the methods directly operates on the encoder's representations that are eventually used for downstream tasks, and hence of interest for a method-independent general definition of memorization. We address this challenge by identifying training data *augmentations* and *alignment*, *i.e.,* similarity in representations over different augmentations of the same training data point, as common elements over all SSL methods. To define memorization of a data point, we consider the difference in alignment of representations for its augmented views produced by encoders that were trained on this point and encoders that were not.

We empirically analyze memorization based on this definition over multiple datasets, encoder architectures, and SSL training methods including contrastive and non-contrastive approaches. Our results highlight that even though SSL relies on large datasets and strong augmentations which are known as regularization techniques against overfitting in supervised learning, a significant fraction of data points still experiences high memorization in SSL. Additionally, while the training process of SSL is substantially different from supervised learning, and while no class labels exist that explicitly make data points "outliers", we still observe that atypical data points experience higher levels of memorization than typical ones, a result similar to the supervised setting (Feldman & Zhang, 2020). We demonstrate this effect visually in Figure 1. Yet, we also find that while different SSL methods and encoder architectures exhibit high memorization on a similar set of data points, the data points that are memorized in supervised learning differ substantially.

Finally, we turn to the question: *why do SSL models memorize?* Our analysis reveals that, in a similar vein to supervised learning, also in SSL, memorization improves downstream generalization. The key insight from our empirical evaluation, and the main difference to supervised learning, is that this holds over various downstream tasks, *i.e.,* the encoder memorizing data points from one distribution yields better downstream generalization for another distribution. We even observe this effect on non-classification downstream tasks, such as semantic segmentation. This highlights that memorization improves SSL's general success on various downstream tasks.

In summary, we make the following contributions.

- We propose a formal definition of memorization for SSL encoders (`SSLMem`) that is independent of the training method, its concrete training loss, and that operates directly on the representations.

- We empirically evaluate our definition in practice and find that over different architectures and training methods in SSL, there is significant memorization, especially of atypical data points. While the points with the highest memorization scores align between different SSL training methods,

especially when they share the same architecture, they differ more substantially between SSL and supervised learning.

- We show that SSL memorization in the encoder increases the downstream generalization over different downstream data distributions and tasks.

## 2 BACKGROUND AND RELATED WORK

**SSL.** Self-Supervised Learning (SSL) trains encoder models to transform unlabeled inputs into useful representations which enable sample-efficient learning of multiple downstream tasks (Bengio et al., 2013). Recently, many methods were proposed for learning from large amount of unlabeled data in the vision domain (Chen et al., 2020; Chen & He, 2021; Caron et al., 2021; Bardes et al., 2022; He et al., 2022). Since our work is focused on providing a universal definition of memorization in SSL, we consider different approaches that rely on three distinct learning objectives. **Contrastive learning** was pioneered by work on SimCLR (Chen et al., 2020) where one trains encoders such that augmented views of the same input, also called positive pairs, obtain representations close to each other while representations for dissimilar inputs (negative pairs) are repelled from each other. The key properties of contrastive learning with respect to representations are (1) *alignment* (closeness) of the representations from positive pairs, and (2) sufficient *class center separation* (divergence) (Huang et al., 2023). The foundations for **non-contrastive learning** (Pokle et al., 2022) were laid by SimSiam (Chen & He, 2021) which showed that negative samples are unnecessary to avoid trivial solutions, such as encoder collapse, where the same representation is returned for each input. By training with a projection head applied to only one of its Siamese encoders and preventing the gradient from propagating through the other encoder branch, SimSiam is able to generate representations with high alignment and class center separation. DINO (Caron et al., 2021) further extended this strategy by minimizing the cross-entropy loss (on latent classes obtained through the training head) instead of negative cosine similarity and decorrelating the two Siamese branches. Finally, **masked autoencoding**, such as canonical MAE (He et al., 2022), trains an asymmetric encoder-decoder architecture instead of two-branched encoders. MAE learns to reconstruct randomly masked patches of an input image. By masking high portions (75%) of inputs, this strategy encourages learning useful features and enables better scalability (faster pre-training and less memory consumption). The distinguishing factor between MAE and other SSL encoders is its reliance on the masking of inputs instead of strong dependence on other data augmentations, such as, random cropping or color jitter.

**Membership Inference Attacks.** The standard approach for measuring how machine learning models leak private information about their training data is through membership inference (Shokri et al., 2017), where an adversary attempts to determine if a particular data point was used to train a given model. *EncoderMI* (Liu et al., 2021) detects encoder membership by observing that alignment scores for training data points are higher than for points not used in training. While we leverage a similar concept to study memorization, we do not narrow our analysis down to quantifying privacy risks of a particular encoder. Instead, we use the concept of memorization to study broader properties of SSL. Above all, we establish how memorization influences the downstream generalization.

**Memorization.** As an important property of learning algorithms and neural networks, memorization has been actively studied in supervised learning (Zhang et al., 2016; Arpit et al., 2017; Chatterjee, 2018; Carlini et al., 2019; 2021; 2022). The fundamental idea to quantify memorization relies on the impact that a single training data point has on the predictions of the resulting models with a larger impact (or increased "hardness" of learning the data point (Arpit et al., 2017; Sadrtdinov et al., 2021)) indicating a higher level of memorization (Feldman, 2020). While memorization has been shown to be important for generalization related properties in supervised learning (Feldman, 2020; Feldman & Zhang, 2020) or the supervised downstream classifiers within transfer learning (Bansal et al., 2020), this aspect has not been studied for SSL. The only work which considers memorization in SSL proposes the concept of Déjà Vu memorization (Meehan et al., 2023) and quantifies how much SSL encoders associate specific views (for example of background crops) with the foreground objects in training images. To assess whether an encoder exhibits Déjà Vu memorization for a given training data point, the framework obtains the representation for a crop of the data point, and compares the representation with representations of labeled data points from a public dataset that has the same distribution as the encoder's training data. If the labels within the $k$ nearest neighbors of the crop in the representation space are highly consistent, the data point is marked as memorized. Since the

core property of SSL is training without labels, the biggest limitation of the Déjà Vu memorization is the assumption about access to labeled data from the same distribution as the training set of the encoder. Additionally, Déjà Vu memorization relies on a particular SSL augmentation, namely cropping. However, not all SSL methods use cropping. Finally, SSL encoders are applied to a myriad of downstream tasks other than classification (*e.g.,* multi-label classification, segmentation, depth detection) where the concept of a single class per input does not exist—rendering this definition of memorization narrow. Our definition of memorization is based on representations, which are output by all SSL encoders, thus it is independent of particular augmentations, downstream tasks, or the availability of auxiliary information, such as class labels.

## 3 TOWARDS FORMALIZING MEMORIZATION

Given the absence of labels in SSL, directly applying definitions of general memorization from supervised learning, such as (Feldman, 2020), is inadequate. Therefore, we aim at deriving a new definition of memorization suitable for SSL and independent of a specific learning framework (*e.g.,* contrastive learning (Chen et al., 2020) or masked autoencoding (He et al., 2022)).

Our definition leverages a common element over all SSL frameworks, namely data *augmentations* and their *alignment*. Augmentations refer to different views of a data point, generated, for instance, through cropping, masking, or noise addition. Informally speaking, when learning with SSL, the objective is to obtain an encoder that achieves a *low alignment loss* on different augmented views of a training data point, *i.e.,* an encoder that returns very similar representations on the training data point and its augmentations. Note that different SSL methods, in addition to alignment, optimize implicitly (He et al., 2022; Zhang et al., 2022) or explicitly (Chen et al., 2020) for other objectives, such as uniformity. Yet, given that these are not properties of an individual data point but rather of the overall representation space, influenced by multiple data points, we do not include them into the definition of our per-data point memorization.[1] Instead, we use representation alignment between different augmented views of a data point to detect memorization. More concretely, we consider a data point as having a high level of memorization by an encoder $f$ if its alignment is significantly higher on $f$ than on encoder $g$ that was not trained with the considered data point. In the following, we will formalize this intuition and propose our novel definition for memorization in SSL (`SSLMem`).

### 3.1 PRELIMINARIES AND PROBLEM SETUP

We present a formal model of SSL learning methods as well as concepts that are relevant to defining memorization. In doing so, we leverage several of the main ideas proposed by recent theoretical work on SSL (Parulekar et al., 2023; Huang et al., 2023; Wang et al., 2022). Let $f : \mathbb{R}^n \to \mathbb{R}^d$ be an encoder trained using an SSL learning algorithm $\mathcal{A}$ on an unlabeled training dataset $S = \{x_i\}_{i=1}^m$. We assume randomness in the training algorithm, *e.g.,* random weight initializations, such that the final trained encoder $f$ is from a class of possible encoders $\mathcal{F}$. For each data point $x$, we define an augmentation set $\text{Aug}(x) = \{a(x) | a \in \text{Aug}\}$ where $a$ is an augmentation, *i.e.,* a transformation from $\mathbb{R}^n \to \mathbb{R}^n$, and Aug is the set of all possible augmentations. $f(x)$ denotes an output representation of encoder $f$ for the data point $x$. We measure the distance between representations of two different augmentations $x'$ and $x''$ of $x$ with a metric $d$, *e.g.,* the $\ell_2$ distance, and define alignment loss over the representations as

$$\mathcal{L}_{\text{align}}(f, x) = \mathop{\mathbb{E}}_{x', x'' \sim \text{Aug}(x)} [d\left(f(x'), f(x'')\right)]. \tag{1}$$

A standard downstream task for a trained SSL encoder $f$ is classification, where a linear layer $G_f$ is trained to map from the representation space produced by $f$ to labels (this form of evaluation is also referred to as *linear probing*). With respect to classification, the generalization error of encoder $f$ is defined in terms of the error that classifier $G_f$ achieves. The main connection between a low alignment loss of $f$ over the augmentation set of each training data point and the error of $G_f$ on downstream tasks is based on the overlap of augmentation sets. Considering two data points $x_1, x_2$ from the same downstream class, it is likely that they will have overlapping augmentation sets (*i.e.,* $\exists a_1, a_2 \in \text{Aug}$ s.t. $a_1(x_1) = a_2(x_2)$) (Huang et al., 2023). When the alignment loss decreases, the difference between $\text{Aug}(x_1)$ and $\text{Aug}(x_2)$ decreases. This will lead to $d(f(x_1), f(x_2))$ also

---

[1]We provide a formal discussion on the fact that alignment is part of all SSL methods, and what other optimization objectives different methods exhibit in Appendix D.2

decreasing by the triangle inequality.[2] Hence $x_1, x_2$ will obtain similar representations, facilitating $G_f$ in assigning them the same class label (Huang et al., 2023).

## 3.2 ALIGNMENT AND MEMORIZATION IN SSL

Our definition for memorization in SSL follows the *leave-one-out* definition of memorization from supervised learning (Feldman, 2020) but instead of focusing on the model behavior w.r.t. ground truth labels (which do not exist in SSL), it is based on the alignment loss (1) of training data points. Consider a single data point $x$ from dataset $S$ and encoders $f \in \mathcal{F}, g \in \mathcal{G}$ trained with SSL algorithm $\mathcal{A}$ on $S, S \setminus x$ (dataset $S$ with $x$ removed), respectively. We then define the memorization score $m$ with SSLMem on $x$ as

$$m(x) = \underset{g \sim \mathcal{A}(S \setminus x)}{\mathbb{E}} \underset{x', x'' \sim \text{Aug}(x)}{\mathbb{E}} [d(g(x'), g(x''))] - \underset{f \sim \mathcal{A}(S)}{\mathbb{E}} \underset{x', x'' \sim \text{Aug}(x)}{\mathbb{E}} [d(f(x'), f(x''))]. \quad (2)$$

Here, we take the expectation not only over the set of augmentations of $x$, but also over two different function classes consisting of all possible encoders $f, g$ which can result from the SSL training algorithm. Specifically, these classes are $\mathcal{F} = \mathcal{A}(S)$ and $\mathcal{G} = \mathcal{A}(S \setminus x)$. Intuitively, our definition quantifies how the alignment of representations in $\text{Aug}(x)$ varies between encoders $f$ and $g$. Following the intuition from Feldman & Zhang (2020), our memorization score is higher for a data point $x$ if the alignment changes significantly between $f$ and $g$, *i.e.,* based on whether $x$ was used for training or not. Importantly, alignment and memorization report different concepts: the former is a direct property of a given encoder whereas the latter is a result of the relative comparison between different families of encoders. In particular, low alignment loss does not necessarily correspond to high memorization, which we show in Figure 2a (the bottom left corner). To understand why this holds, consider a candidate data point $x$ included in the training set of encoders $f \in \mathcal{F}$ but not in the one of encoders $g \in \mathcal{G}$. $f$ can have a low alignment loss but also low memorization on $x$. This happens if $g$ has an equally low alignment loss on $x$ as $f$, for example, because $x$ is easy to learn or similar to other examples in $g$'s training set. Note that we subtract the term for encoders $f \in \mathcal{F}$ from the term for $g \in \mathcal{G}$ to obtain a positive memorization score with the expectation that encoders $g$ which are trained without $x$ usually have a higher alignment loss on $x$ than encoders $f$. We provide further theoretical analysis of alignment and memorization for SSL in Appendix D.

## 4 EXPERIMENTAL EVALUATION

To experimentally approximate the memorization score, SSLMem, from Equation (2), we consider averaging over five random augmentations. We divide the training set $S$ into three disjoint partitions. For example, in CIFAR10, we use 80% of the train data, *i.e.,* 40000 samples as shared training data $S_S$ between encoders $f$ and $g$. The next 10% of samples, *i.e.,* 5000 are used as candidates $S_C$ to evaluate memorization. We add those to the training data of $f$ only, and the remaining 10%, which is another 5000 samples, are used as an independent set $S_I$, on which we do not train $f$ but only $g$. We also use additional extra set $S_E$ with 5000 samples from the test set, which are data points not used for training of either $f$ or $g$. Thus encoder $f$ is trained on $S_S \cup S_C$, whereas $g$ is trained on $S_S \cup S_I$. We measure the memorization on the candidates $S_C$ and report their average memorization scores as an aggregate metric. To provide more fine-grained qualitative insights into the memorized samples, we additionally report overviews on the per-data point distributions and zoom into the points that experience the highest memorization. We use 50000 data points as training samples for CIFAR10, SVHN, and STL10 and 100000 for ImageNet. We set the batch size to 1024 for all our experiments and train for 600 epochs on CIFAR10, SVHN, and STL10, and for 300 epochs on ImageNet. As a distance metric to measure representation alignment, we use the $\ell_2$ distance. To be able to compare memorization between different SSL methods, we normalize the resulting memorization scores to a range between -1 and 1. A memorization score of 0 denotes no memorization, +1 is the strongest memorization effect on encoder $f$, and -1 strongest memorization on $g$. We repeat all experiments with three independent seeds and report the average SSLMem memorization and standard deviation. Our full experimental setup is depicted in Appendix A.

---

[2]We assume that after training $\mathcal{L}_{\text{align}}(f, x_1), \mathcal{L}_{\text{align}}(f, x_2) \leq c$. When considering the region $\text{Aug}(x_1) \cap \text{Aug}(x_2)$, we can find a point $y$ in this region so that both of $d(f(x_1), f(y))$ and $d(f(x_2), f(y))$ are $\leq c$. Now the triangle inequality can be used to obtain $d(f(x_1), f(x_2)) \leq 2c$.

Table 1: **Higher memorization for more performant encoders.** We present the average memorization score over the 5000 candidates $S_C$ (SSLMem) and the linear probing accuracy (one layer for classification trained on top of the representations for the respective datasets, Acc.) over various datasets, encoder architectures, and SSL training methods. SimCLR and DINO are trained using ResNet50. MAE and DINO are also trained with the ViT architecture. We use ViT-Tiny for all datasets, apart from ImageNet, for which we use ViT-base.

| | | CIFAR10 | | SVHN | | STL10 | | ImageNet | |
|---|---|---|---|---|---|---|---|---|---|
| method | Model | SSLMem | Acc. (%) | SSLMem | Acc. (%) | SSLMem | Acc. (%) | SSLMem | Acc. (%) |
| MAE | VIT | $0.307 \pm 0.013$ | $67.40\% \pm 1.10\%$ | $0.311 \pm 0.009$ | $68.52\% \pm 1.02\%$ | $0.284 \pm 0.011$ | $62.11\% \pm 0.95\%$ | $0.271 \pm 0.004$ | $60.43\% \pm 1.18\%$ |
| DINO | VIT | $0.334 \pm 0.010$ | $76.12\% \pm 0.79\%$ | $0.356 \pm 0.011$ | $82.26\% \pm 1.48\%$ | $0.321 \pm 0.008$ | $73.88\% \pm 0.85\%$ | $0.309 \pm 0.015$ | $68.21\% \pm 1.55\%$ |
| DINO | ResNet50 | $0.327 \pm 0.009$ | $75.39\% \pm 1.15\%$ | $0.350 \pm 0.014$ | $80.69\% \pm 0.94\%$ | $0.319 \pm 0.007$ | $73.02\% \pm 1.92\%$ | $0.311 \pm 0.012$ | $68.44\% \pm 0.61\%$ |
| SimCLR | ResNet50 | $0.339 \pm 0.011$ | $77.12\% \pm 1.42\%$ | $0.357 \pm 0.008$ | $82.30\% \pm 1.31\%$ | $0.321 \pm 0.009$ | $74.22\% \pm 1.66\%$ | $0.301 \pm 0.011$ | $66.12\% \pm 1.23\%$ |

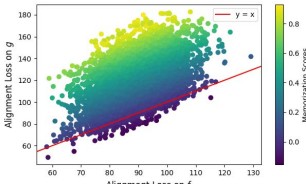

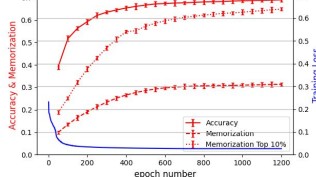

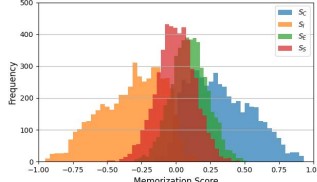

(a) Model Alignment Loss computed according to (1) vs. Memorization.

(b) Connection between training loss, downstream accuracy, and memorization scores.

(c) Comparison between memorization scores for data subsets $S_C$, $S_I$, $S_E$, and $S_S$.

Figure 2: **Insights into our memorization score.** We train an MAE with VIT-tiny on CIFAR10. (a) We plot the alignment loss, computed with the $\ell_2$ distance, of the candidates (with respect to their augmentation) on encoder $f$ and encoder $g$. The color coding indicates the memorization score with higher scores indicating higher memorization. The lowest alignment loss on $f$ does not yield the highest memorization score, and high memorization can occur at a wide range of alignment loss values for $f$. (b) Training loss, downstream accuracy, and memorization over the course of training highlight that memorization is not just an effect of increasing/decreasing accuracy: while loss and accuracy stagnate after a few hundred epochs, memorization increases. (c) We report the memorization scores for 5000 data points from each subset $S_C$, $S_I$, $S_E$, and $S_S$. The encoders exhibit memorization indicated by significantly higher (lower) scores for $S_C$ ($S_I$) compared to $S_S$ or $S_E$.

## 4.1 Memorization over Different Architectures, SSL methods and Datasets

We assess memorization over different encoder architectures, SSL training methods, and datasets and report the results in Table 1. Our analysis of memorization shows a correlation between downstream task accuracy and the average memorization score. This trend holds for SimCLR on CIFAR10, SVHN, and STL10, and for DINO on ImageNet, where these methods achieve the highest accuracy and the biggest average SSLMem memorization score, while MAE exhibits the lowest scores on both measures across all four considered datasets. We present additional results on the impact of type and strength of augmentations and the training method on memorization in Appendix B.1. Overall, greater average SSLMem memorization appears to be associated with superior downstream performance. Yet, as we illustrate in Figure 2b alignment and accuracy are distinct metrics. Training loss and accuracy plateau after a few hundred epochs, but memorization continues increasing with longer training. This holds both over the entire candidate-set, and especially for the 10% most memorized samples—highlighting that more epochs lead to higher memorization. This insight decouples the the measures of accuracy and memorization in terms of training dynamics.

## 4.2 Insights on the Memorization Score

The results shown in Figure 2c demonstrate that our memorization score behaves as expected. Memorization significantly increase above 0 for the candidate samples $S_C$ used during training of encoder $f$ for which we want to capture memorization, significantly decrease below 0 for the independent samples $S_I$ used for training of encoder $g$, while remaining around 0 for the shared samples $S_S$ or extra data points $S_E$ not seen during training. We formally verify that data points from $S_C$ ($S_I$) have statistically significantly higher (lower) SSLMem memorization scores $m$ than those from $S_S$ and $S_E$.

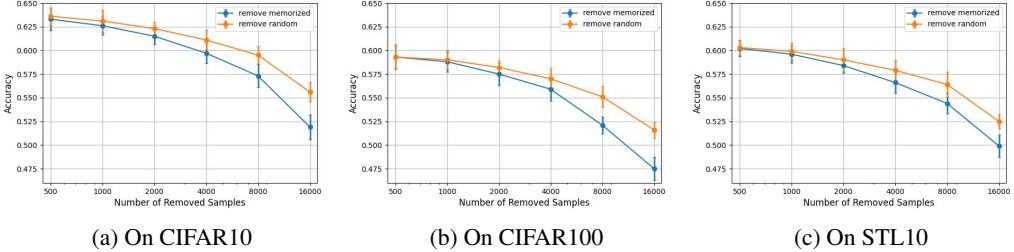

| (a) On CIFAR10 | (b) On CIFAR100 | (c) On STL10 |

Figure 3: **The influence of memorization on downstream generalization (CIFAR10).** We train an MAE model based on the VIT-tiny architecture on CIFAR10 and remove [500, 1k, 2k, 4k, 8k, 16k] most memorized vs. random data points from the encoder's training data. We measure downstream accuracy through linear probing on CIFAR10, CIFAR100, and STL10. The removal of memorized data points harms accuracy over all downstream tasks more than the removal of random data points.

Table 3: **Evaluating the effect of memorization on a semantic segmentation downstream task.**

|  | Without removing | Removing 10000 | | Removing 20000 | |
|  |  | Memorized | Random | Memorized | Random |
| --- | --- | --- | --- | --- | --- |
| mIoU | 45.4 | 44.8 | 45.1 | 43.8 | 44.4 |
| Acc. (%) | $69.89\% \pm 0.84\%$ | $68.33\% \pm 0.92\%$ | $68.91\% \pm 0.77\%$ | $66.51\% \pm 1.03\%$ | $67.58\% \pm 0.82\%$ |

The mean memorization scores are as follows for each of the subsets: 0.30723 for $S_C$, -0.00136 for $S_S$, 0.09958 for $S_E$, and -0.31182 for $S_I$. Using a t-test with 5000 memorization scores per each data subset, we test the null hypothesis $\mathcal{H}_0 := m(S_C) \leq m(S_S)$. Rejecting this hypothesis (p-value $< 0.01$) indicates the memorization $m$ is significantly higher for points in $S_C$ than for

Table 2: **Results of statistical t-tests.**

| Null Hypothesis | p-value | effect size |
| --- | --- | --- |
| $\mathcal{H}_0 := m(S_C) \leq m(S_S)$ | 0 | 86.82 |
| $\mathcal{H}_0 := m(S_C) \leq m(S_E)$ | 0 | 60.44 |
| $\mathcal{H}_0 := m(S_S) \leq m(S_I)$ | 0 | 85.48 |
| $\mathcal{H}_0 := m(S_E) \leq m(S_I)$ | 0 | 113.56 |

points in $S_S$. Our results reject $\mathcal{H}_0$ with a small p-value near 0 and a large effect size of 86.82, which indicates that observed difference is not only statistically significant but also meaningful. We show the results of the statistical tests for all the considered data subsets in Table 2. They support the claim that $S_C$ ($S_I$) is substantially more (less) memorized than $S_S$ and $S_E$. We also observe that memorization scores for both $S_E$ and $S_S$ have their peaks close to 0. There is a difference in the mean scores between $S_E$ than $S_S$ since data points from $S_E$ are not seen during training of neither $f$ nor $g$ while data points from $S_S$ are used for the training of both $f$ and $g$. We present further analysis in Appendix C.

### 4.3 MEMORIZED DATA POINTS

Additionally, we analyze what types of data points are memorized. In Figure 1, we already showed visually that, similar to supervised learning, atypical examples experience a higher memorization in SSL than standard data points. Additionally, we show in Figure 9 and Table 13 in Appendix B.4 that SSL and supervised learning differ notably in the data points that they assign the highest memorization scores to while the SSL setups memorize in a more similar way. Especially, the SSL setups that share the same training method or the same encoder architecture are most consistent.

### 4.4 MEMORIZATION IN SSL IS REQUIRED FOR DOWNSTREAM GENERALIZATION

**Classification.** We empirically analyze how memorization impacts downstream generalization to classification tasks by removing the most memorized data points from the training data of an encoder and assessing its linear probing accuracy on downstream tasks. More concretely, we train $f$ and $g$ encoders with MAE using the ViT-tiny architecture on disjoint 25k data points from the CIFAR10 training dataset. Then, we measure the memorization scores over encoder $f$ and remove the [500, 1k, 2k, 4k, 8k, 16k] data points with the highest memorization scores from training. We do the same for randomly chosen [500, 1k, 2k, 4k, 8k, 16k] data points from encoder $f$ and compare downstream accuracy on multiple downstream tasks through linear probing on both these setups. Our results in Figure 3 highlight that removing the memorized data points harms downstream accuracy stronger

than removing random data points. This does not only hold when the SSL encoder was trained with the same dataset as the downstream task but also when the downstream task comes from a different distribution (STL10) or has a different number of classes (CIFAR100). In Appendix B.2 and Appendix B.5, we show that this trend holds over different training and downstream datasets.

**Semantic Segmentation.** In a similar evaluation setup, for semantic segmentation downstream tasks, we pre-train a ViT-base with MAE on ImageNet, evaluate memorization, and remove the top [10k, 20k] memorized vs. random data points from the encoder's pre-training data. We end-to-end fine-tune the resulting encoders with UperNet (Xiao et al., 2018) on the ADE20K dataset. We measure downstream accuracy on ImageNet for the fine-tuned encoder through linear probing and the semantic segmentation performance with the mean Intersection of Union (mIoU). Removing memorized samples from pre-training harms downstream performance on the semantic segmentation more than removing random samples, even after an independent end-to-end fine-tuning. In Appendix B.2, we show similar results for the downstream task of *depth estimation*.

The observations on the interplay between memorization in SSL and its impact on the performance on diverse downstream tasks is a core result of this work, highlighting the importance of memorization for generalization of encoders, beyond the encoders own training distribution. To further validate the result, we investigate the effect of limiting alignment during the encoder training on both memorization and downstream accuracy. With an alignment limited through regularization, the difference between encoders $f$ and $g$ on data points that were in the training set of $f$ but not of $g$ should decrease, which would result in a decreased memorization score. To implement this intuition, we extend the loss function during training with an additional term as:

$$\mathcal{L}_{\text{total}}(f, x) = (1 - \lambda)\mathcal{L}_{SSL}(f, x) - \lambda \mathop{\mathbb{E}}_{x', x'' \sim \text{Aug}(x)} [d\left(f(x'), f(x'')\right)] \tag{3}$$

The additional term $\mathop{\mathbb{E}}_{x', x'' \sim \text{Aug}(x)} [d\left(f(x'), f(x'')\right)]$ directly penalizes representations of a data point and its augmentation set for being too close. The parameter $\lambda$ quantifies regularization strength with smaller values representing a weaker regularization. Note that this regularization term does not directly invert the effect of SSL training which does not optimize directly on the representation space.

For example, MAE training loss is calculated on the decoder output space for the reconstructed samples. Other SSL methods, such as SimSiam and SimCLR, map representations to the output space of the projection head where the loss is applied. In contrast, our regularization term operates directly on the representations themselves, in order to ensure an explicit control of the alignment.

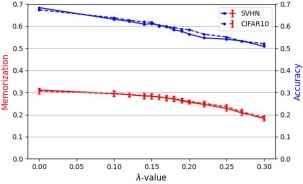

We evaluate the effect of the additional loss term in Figure 4 for a ViT-tiny model, trained with MAE on CIFAR10 (solid lines) and SVHN (dashed lines) under different values for $\lambda$. In the implementation, we instantiate $d$ with the $\ell_2$ distance and take the expectations over two random augmentations of the original data point. We calculate both loss terms over a whole mini-batch, not a single data point, with

Figure 4: **Limiting memorization harms downstream accuracy.**

a mini-batch size of 256. Our results demonstrate that increasing regularization strength (higher $\lambda$) reduces model memorization. Concurrently, downstream accuracy from linear probing also decreases. This aligns with previous work showing that better alignment enables better generalization on downstream tasks (Huang et al., 2023). We expand upon these analyses by highlighting that limiting memorization capabilities negatively impacts encoder performance.

## 4.5 COMPARISON TO PRIOR WORK

The Déjà Vu memorization and our memorization score capture different phenomena and measure memorization in distinct ways. Déjà Vu memorization reports the fraction of data points classified as memorized based on label consistency with nearby points from the labeled dataset. Our method measures per-point memorization scores and reports the average score over the candidates. Despite the divergent methodologies underpinning each memorization score, we nonetheless endeavor to analyze whether the two scores show similar trends. We report our results in Table 4. We observe that both memorization scores are higher on CIFAR10 than on ImageNet. We reason that the memorization is easier on CIFAR10 due to lower-dimensionality of CIFAR10 than ImageNet, smaller number of

Table 4: **Comparing our average `SSLMem` memorization score with Déjà Vu memorization.** We train different model types and measure the memorization with our framework (`SSLMem`) and the Déjà Vu memorization (Déjà Vu Mem.) (Meehan et al., 2023).

| Model | CIFAR10 | | | ImageNet | | |
|---|---|---|---|---|---|---|
| | SSLMem | Deja Vu Mem. | Acc (%) | SSLMem | Deja Vu Mem. | Acc (%) |
| MAE | $0.307 \pm 0.013$ | $27.36\% \pm 1.50\%$ | $67.40\% \pm 1.10\%$ | $0.271 \pm 0.004$ | $21.30\% \pm 0.31\%$ | $60.43\% \pm 1.18\%$ |
| DINO | $0.334 \pm 0.010$ | $25.52\% \pm 0.98\%$ | $76.12\% \pm 0.79\%$ | $0.309 \pm 0.015$ | $20.08\% \pm 0.62\%$ | $68.21\% \pm 1.55\%$ |
| VICReg | $0.334 \pm 0.012$ | $25.20\% \pm 0.49\%$ | $76.46\% \pm 0.94\%$ | $0.311 \pm 0.010$ | $20.62\% \pm 1.11\%$ | $69.05\% \pm 1.08\%$ |

training data points, and using encoders with the same number of parameters for both datasets. We observe a key divergence between the two memorization scores on MAE encoders. Specifically, Déjà Vu produces much higher memorization scores for MAE compared to other SSL methods. In contrast, our `SSLMem` memorization score yields lower scores for MAE than for other SSL methods. We hypothesize that this is due to MAE's training approach which heavily masks input patches, and thereby creates a strong correlation between some background fragments and a foreground object which can be exploited by Déjà Vu. The other SSL methods rely on additional or different augmentations that cannot be so effectively leveraged by Déjà Vu. Our analyses indicate that the specific augmentations employed do not show a statistically significant effect on our `SSLMem`.

## 4.6 DIFFERENTIAL PRIVACY

Differential privacy (Dwork, 2006) provides mathematically rigorous protections against privacy leakage. This framework formalizes the intuition that any individual data point should have negligible influence on the analysis of an entire dataset. In machine learning, differential privacy is often implemented through the DP-SGD algorithm (Abadi et al., 2016), which introduces controlled noise during training and bounds the influence of each individual data point on model updates. However, DP-SGD has a limited compatibility with many

Table 5: **Effect of differential privacy.**

| $\varepsilon$ | SSLMem | Acc. (%) |
|---|---|---|
| $\infty$ | $0.307 \pm 0.013$ | $69.40\% \pm 1.12\%$ |
| 20 | $0.182 \pm 0.009$ | $54.22\% \pm 0.98\%$ |
| 8 | $0.107 \pm 0.012$ | $33.66\% \pm 1.76\%$ |

self-supervised learning paradigms, wherein individual samples influence model updates across their entire mini-batch. Nonetheless, Yu et al. (2023) recently proposed a differentially private training framework for MAE encoders. Our analysis shows that indeed encoders trained with DP-SGD demonstrate reduced memorization. To assess its effect on our `SSLMem` memorization score, we train SSL encoders with the framework by Yu et al. (2023) on MAE and the ViT-tiny architecture. We train for 1000 epochs on CIFAR10 using all default parameters from that work (Yu et al., 2023) apart from their large mini-batch sizes that do not match the limited availability of data in CIFAR10. To report a standard, non-private baseline, *i.e.,* $\varepsilon = \infty$, we train a standard MAE. Our results in Table 5 show that whilst differential privacy indeed reduces memorization depending on the privacy parameter $\varepsilon$, it also substantially reduces downstream accuracy. This can be seen as another indicator that learning abilities in SSL suffer without memorization.

## 5 CONCLUSION

SSL has emerged as a dominant paradigm for training encoders, since it can leverage the abundant amounts of available unlabeled data to create high-quality feature extractors. However, despite their unprecedented performance, the memorization property of self-supervised encoders remain unexplored. Due to the lack of labels, a structured assessment of memorization, as commonly done in supervised learning, could not be conducted previously. We close this gap by providing an analysis of encoder memorization in SSL. Therefore, we first propose a definition for memorization based on augmentations and alignment of positive pairs—the common elements throughout all SSL methods. Our `SSLMem` definition reflects SSL's lack of ground-truth labels, generalizes across different encoder architectures and SSL training algorithms, and is independent of any downstream task. Crucially, we demonstrate that self-supervised encoders do memorize training data points, especially atypical examples. Further, we empirically show that memorization improves generalization on various downstream tasks, even beyond the encoder's pre-training data and its distribution, and beyond simple single label classification tasks. This establishes memorization as a key property of self-supervised feature learning.

ACKNOWLEDGMENTS

We would like to acknowledge our sponsors, who support our research with financial and in-kind contributions: Amazon, Apple, CIFAR through the Canada CIFAR AI Chair, DARPA through the GARD project, Intel, Meta, NSERC through the Discovery Grant, the Ontario Early Researcher Award, and the Sloan Foundation. Resources used in preparing this research were provided, in part, by the Province of Ontario, the Government of Canada through CIFAR, and companies sponsoring the Vector Institute.

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

## A    EXPERIMENTAL SETUP

We validate our algorithms mainly on four state-of-art SSL encoders: MAE (He et al., 2022), SimCLR (Chen et al., 2020), DINO (Caron et al., 2021), and VicReg (Bardes et al., 2022). We train these encoders for 300 epochs with ImageNet ILSVRC-2012 (Russakovsky et al., 2015) and 600 epoch with CIFAR10 (Krizhevsky et al., 2009), CIFAR100 (Krizhevsky et al., 2009) , SVHN (Netzer et al., 2011), and STL10(Coates et al., 2011). All other settings for model training and evaluating (linear-probing) are shown in Table 6. The ImageNet and STL10 based encoders are trained on a server with 2 NVIDIA-A100 GPUs. CIFAR10, CIFAR100, and SVHN-based encoders and all linear probing evaluation are performed on a 4090 GPU server with an Intel 13700K processor and 64G RAM. To measure memorization, we divide the datasets as follows: For CIFAR10, CIFAR100, SVHN, and STL10 dataset, we use 40000 shared training samples as $S_S$ and 2 sets of 5000 non-overlapping training samples as $S_C$ and $S_I$. For ImageNet, $S_S$ comprises 85000 samples and $S_C$ and $S_I$ comprise again 5000 samples each.

**Normalization.**    We normalize the representations output by encoders $f$ and $g$ in the $\ell_2$ norm. Then, we calculate the differences in alignment loss per data sample $x$ over both encoders. Afterwards, we normalize these differences by dividing them by the range (largest minus smallest difference), and report the memorization score as the average of the resulting scores over all data points in $S_C$.

**Semantic Segmentation Setup.**    To evaluate the effect of our memorization on semantic segmentation, we end-to-end fine-tune our ImageNet-based MAE encoders (ViT-base) on the ADE20K (Zhou et al., 2019) dataset with UperNet (Xiao et al., 2018) for semantic segmentation. We perform 100 epochs of fine-tuning with a batch size of 16. The learning rate follows the "poly" learning rate schedule with a initial learning rate of 0.02. The relative position bias (Raffel et al., 2020) is only applied during end-to-end fine-tuning.

**Depth Estimation Setup.**    In a similar vein to the semantic segmentation, for depth estimation experiment, we end-to-end fine-tune our ImageNet-based MAE encoders (ViT-base) with a UNet convolutional neural network (Ronneberger et al., 2015) on the NYU-Depth v2 (Nathan Silberman & Fergus, 2012). We report the quality of the depth estimation through the Root Mean Square Error (RMSE), which is defined as:

$$\sqrt{\frac{1}{n}\sum_{p=1}^{n}(y_p - \hat{y_p})^2} \tag{4}$$

where $y_p$ is the true depth from the NYU-Depth v2 dataset and $\hat{y_p}$ is the predicted depth from the model. The smaller the RMSE, the better the performance of the model.

**Measuring Memorization.**    We calculate the memorization score on the full representations returned by the encoders. Especially, for the ViT-based experiments, we concatenate the patch-based representations into one representation vector. This yields the following dimensionalties for ViT-tiny: 192x257 = 49344, for ViT-base: 197*768 = 151296, for ResNet50: 49*2048 = 100352. Note that particular downstream tasks with the ViT encoders use different parts of the representations. For example, for classification, only the representation of the CLS-token (the first of the 257 outputs) is used. For semantic segmentation, only outputs 2-257 are used. Even though it increases compute time, we decided to compute the memorization score over the entire returned representation to make our score independent of the downstream task. As a consequence of the significant difference in output dimensionality, and the fact that we calculate alignment loss with the $\ell_2$ distance,

Table 6: **Experimental Setup.** We provide details on our setup for encoder training and evaluation.

| | Model Training | | | | Linear Probing | | | |
|---|---|---|---|---|---|---|---|---|
| | MAE | SimCLR | DINO | VicReg | MAE | SimCLR | DINO | VicRef |
| Training Epoch (Imagenet / others) | 300 / 600 | 300 / 600 | 300 / 600 | 300 / 600 | 45 / 90 | 45 / 90 | 45 / 90 | 45 / 90 |
| Warm-up Epoch (Imagenet / others) | 30 / 60 | 30 / 60 | 30 / 60 | 30 / 60 | 5 / 10 | 5 / 10 | 5 / 10 | 5 / 10 |
| Batch Size | 2048 | 4096 | 1024 | 256 | 4096 | 4096 | 4096 | 4096 |
| Optimizer | AdamW | LARS | AdamW | SGD | LARS | LARS | LARS | LARS |
| Learning rate | 1.2e-3 | 4.8 | 2e-3 | 3e-3 | 1.6 | 4.8 | 1.6 | 1.6 |
| Learning rate Schedule | Cos. Decay | Cos. Decay | Cos. Decay | Cos. Decay | Cos. Decay | Cos. Decay | Cos. Decay | Cos. Decay |

[1] the format for epoch number is ImageNet / Other

Table 7: **Impact of the fraction of overlap between $f$ and $g$.** We repeated experiments from Table 1 with ResNet50 trained with SimCLR on CIFAR10 with different splits for the overlap (70% overlap, and 85% overlap). For the best comparability, we made sure to have the same number of training data points over all setups (45k).

| $S_S$, and | $S_C$ | $S_I$ | SSLMem | Acc. (%) |
|---|---|---|---|---|
| 35k (70%) | 10k | 10k | $0.325 \pm 0.008$ | $77.95\% \pm 1.23\%$ |
| 40k (80%) | 5k | 5k | $0.339 \pm 0.011$ | $77.12\% \pm 1.42\%$ |
| 42.5 (85%) | 2.5k | 2.5k | $0.337 \pm 0.010$ | $76.84\% \pm 0.85\%$ |

## A.1 EXPERIMENTALLY APPROXIMATING OUR MEMORIZATION SCORE

A completely faithful assessment of our definition of memorization (Equation (2)) would involve, per data point, training multiple encoders with and without this data point and evaluating their representations. Given the large number of parameters and the high number of training epochs required to train in SSL, this is computationally prohibitive. This suggests that, for our experimentation, we have to approximate the memorization score. There are multiple ways to do so with their own advantages and drawbacks. We present the possibility in the following and motivate the choice of our approximation.

**Disjoint subsets between $f$ and $g$.** In as similar vein to Meehan et al. (2023), we could train $f$ and $g$ on completely disjoint subsets of the original training dataset (*e.g.*, 25k+25k in the CIFAR10 case). Yet, in this setup, given that the two encoders' training data differs in all data points, it becomes increasingly hard to attribute the difference in their behavior to individual data points. This motivates our choice to have a joint training set $S_S$ between $f$ and $g$ and make them differ only in a subset of samples. Ideally, this subset would be as small as possible to more faithfully assess the impact of each individual data point. However, choosing smaller subsets leaves us with less samples to evaluate. To address this trade-off, we decided to make $f$ and $g$ overlap in 80% and differ in 10% of their data sets' initial size, and take this 10% data only used for $f$ as candidates. We carried out additional experiments to showcase that the memorization score does not change with higher overlapping ratios (85%) but decreases for smaller ratios (70%) in Table 7. Thus, the ratios below 80% do not provide us with a sufficiently precise measure of memorization and that our choice of 80% is sufficient to well approximate the metric while being computationally efficient and allowing to assess the largest possible number of training data points at the same time.

**Removing or replacing data points in $g$.** After deciding in how many data points $f$ and $g$ should differ, the next choice is regarding how to modify the training data of $g$. Our definition of memorization indicates that the candidates should be removed without replacement in $g$. This enables to clearly measure their effect on training without having potentially different data point interfere. However, we empirically observed that removing 10% of the training data leaves $g$ with a generally worse alignment than $f$. This would skew the memorization score (because the alignment loss of $g$ would be generally higher). As a solution, we decided not to simply remove the candidates for training $g$, but to replace them with an independent data subset $S_I$ of same size from the same distribution.

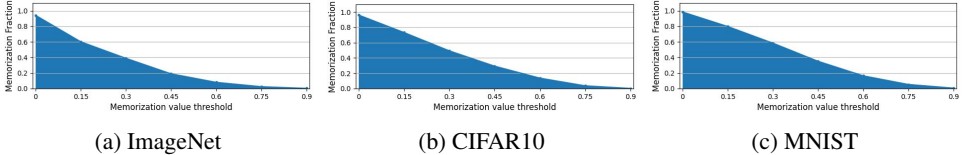

| (a) ImageNet | (b) CIFAR10 | (c) MNIST |

Figure 5: **Influence of the memorization threshold.** Using the MAE-base model, we depict what fraction of data points from the respective candidate dataset would be classified as memorized by our definition when choosing the memorization threshold according to the number depicted on the x-axis.

Table 8: **Impact of measuring memorization with different augmentations than the ones used during training.** We train a ResNet50 on CIFAR10 with SimCLR and measure the memorization score with different augmentations.

| Augmentation for Measurement | Average SSLMem Memorization |
|---|---|
| SimCLR original | $0.339 \pm 0.011$ |
| GaussianNoise (mean=0 and std=0.2) | $0.321 \pm 0.014$ |
| Rotate 90° | $0.308 \pm 0.009$ |
| Rotate 270° | $0.328 \pm 0.011$ |
| ColorDrop 0.25 | $0.298 \pm 0.006$ |

### A.2 THRESHOLDING OF OUR MEMORIZATION SCORE

One important consideration regarding the memorization score concerns the question *When is a data point memorized?* This question could be addressed by setting a threshold on the memorization score that categorizes samples into memorized and non-memorized. Yet, this would indicate that memorization is a binary concept, and it would involve the choice of a threshold. Since this threshold would have to be set arbitrarily (with respect to some desired outcome, like obtaining a certain fraction of memorized data points), we refrain from this choice and rather report the continuous memorization scores. The continuous scale captures nuanced differences in how strongly various data points affect each encoder. Additionally, we show in Figure 5 how the number of samples classified as memorized would change for different memorization value thresholds. This further illustrates that our memorization score forms a continuous spectrum. We present additional structured insights into our memorization score in Appendix C.

## B  ADDITIONAL EXPERIMENTAL RESULTS

### B.1  THE INFLUENCE OF AUGMENTATIONS

We study the impact of the type of augmentations used for training on the average memorization and the linear probing accuracy at the example for SimCLR with a ResNet50 encoder on the CIFAR10 dataset. We show that cropping causes larger average memorization than noise addition or masking. Intuitively, this makes sense given that our memorization score relies on representation alignment where the noised version of a red and a blue car are still far away in input space and, therefore, might result in different representations, whereas a crop of their window or tire might be very similar, resulting in well-aligned representation (Huang et al., 2023). Again, we observe that this is closely related to the linear probing accuracy.

We additionally study the impact of augmentation strength in form of the masking ratio in MAE. We observe that average memorization peaks at a 75% masking ratio, again, aligned with the highest linear probing accuracy.. We present our results in Table 9.

Finally, in Table 8, we depict the impact of measuring memorization with a different set of augmentations than the ones used during training. We experimented with ResNet50 trained on CIFAR10 by SimCLR (77.12% accuracy on the downstream classifier). SimCLR originally uses the following augmentations: RandomResizedCrop(32), RandomHorizontalFlip(p=0.5), ColorJitter(0.4, 0.4, 0.4, 0.1)], p=0.8), and RandomGrayscale(p=0.2). The results indicate that using the same original

Table 9: **The effect of different type and strength of augmentations on memorization.** We train on the CIFAR10 dataset and measure the effect of different augmentation types (for SimCLR) and augmentations strengths (in the form of the masking ratio in MAE) on the average memorization score and linear probing accuracy.

|  | Avg. Mem. | linear probing acc (%) |
|---|---|---|
| crop (only) | $0.322 \pm 0.010$ | $74.51\% \pm 1.38\%$ |
| crop+resize | $0.326 \pm 0.014$ | $75.22\% \pm 0.96\%$ |
| random Gaussian noise | $0.319 \pm 0.006$ | $71.94\% \pm 1.62\%$ |
| random masking (75% MAE) | $0.288 \pm 0.012$ | $63.71\% \pm 1.06\%$ |

(a) Different augmentation types for SimCLR trained with ResNet50.

| masking ratio | Mem.Frac. | linear probing acc (%) |
|---|---|---|
| 50% | $0.283 \pm 0.010$ | $62.09\% \pm 0.43\%$ |
| 75% | $0.307 \pm 0.012$ | $67.40\% \pm 1.10\%$ |
| 80% | $0.300 \pm 0.011$ | $65.06\% \pm 1.35\%$ |
| 90% | $0.249 \pm 0.009$ | $58.77\% \pm 1.26\%$ |

(b) Different augmentation strengths implemented through different masking ratios in MAE with the ViT Tiny architecture.

Table 10: **Evaluating the effect of memorization on a depth estimation.** We pre-train a ViT-base with MAE on ImageNet and remove the top [10k, 20k] memorized vs. random data points. We end-to-end fine-tune the resulting encoders on the NYU-Depth v2 (Nathan Silberman & Fergus, 2012). We report the quality of the depth estimation through the Root Mean Square Error (RMSE).

|  | Without removing | Removing 10000 | | Removing 20000 | |
|---|---|---|---|---|---|
|  |  | Memorized | Random | Memorized | Random |
| RMSE | 0.289 | 0.295 | 0.292 | 0.311 | 0.302 |
| Acc. (%) | $70.22\% \pm 1.15\%$ | $69.10\% \pm 0.88\%$ | $69.61\% \pm 0.98\%$ | $67.31\% \pm 1.36\%$ | $68.28\% \pm 1.02\%$ |

augmentations that were used during training also for measuring memorization yields the highest memorization score, *i.e.,* gives the strongest signal to measure memorization. Yet, the other augmentations' scores are not significantly different, and hence can be used equally to approximate the degree of memorization.

## B.2 LINK BETWEEN MEMORIZATION AND GENERALIZATION

**Classification.** In a similar vein to Figure 3 in the main paper, we repeat the experiment and pretrain the encoder on STL10 Figure 6 and CIFAR100 Figure 7. We remove the top memorized vs. random data points and measure linear probing accuracy on CIFAR10, CIFAR100, and STL10. Our results show that over all datasets, even though they have different numbers of classes, or come from different distributions, it holds that the removal of memorized data points has a more detrimental effect to accuracy than the removal of random points. Results for more fine-grained datasets (ImageNet, Food-101, and Flower102) can be found in Appendix B.5.

**Depth Estimation.** In a similar vein to the segmentation downstream task, we pre-train a ViT-base with MAE on ImageNet, evaluate memorization, and remove the top [10k, 20k] memorized vs. random data points from the encoder's pre-training data. We end-to-end fine-tune the resulting encoders on the NYU-Depth v2 (Nathan Silberman & Fergus, 2012). We measure downstream accuracy on ImageNet for the fine-tuned encoder through linear probing and the quality of the depth estimation through the Root Mean Square Error (RMSE). Smaller RMSE indicates a better depth estimation. Our results in Table 10 highlight that removing memorized samples from pre-training harms downstream performance on the depth estimation more than removing random samples.

## B.3 MEMORIZATION IN SUPERVISED LEARNING AND SSL

**Analyzing supervised models' internal representations with `SSLMem`.** To analyze memorization in supervised learning with our score, we train a ResNet50 in a supervised way on CIFAR10, using the cross-entropy loss. Then, we turn the resulting model in into an encoder by removing the last (classification) layer. We train the supervised model for a numbers of epochs. After 10 epochs, the accuracy of the model roughly matches the linear probing accuracy of the encoder trained with DINO. After 100 epochs of supervised training, the training loss plateaus. We repeat the experiment three times and report the average and standard deviation. The results are reported in Table 11. Our results highlight the supervised models trained until convergence have the highest memorization score,

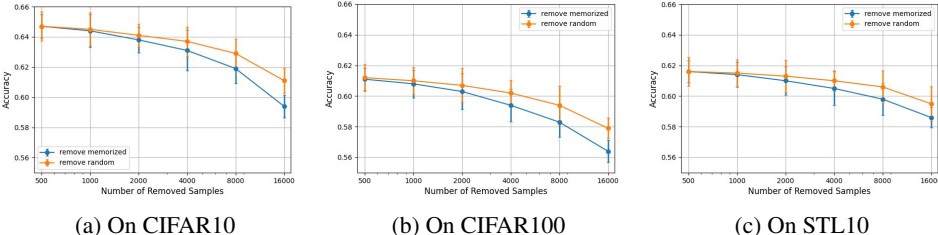

| (a) On CIFAR10 | (b) On CIFAR100 | (c) On STL10 |

Figure 6: **The influence of memorization on downstream generalization (STL10).** We train an MAE model based on the VIT-tiny architecture on STL10 and remove [500, 1k, 2k, 4k, 8k, 16k] most memorized vs. random data points from the encoder's training data. We measure downstream accuracy through linear probing on CIFAR10, CIFAR100, and STL10. The removal of memorized data points harms accuracy over all downstream tasks more than the removal of random data points.

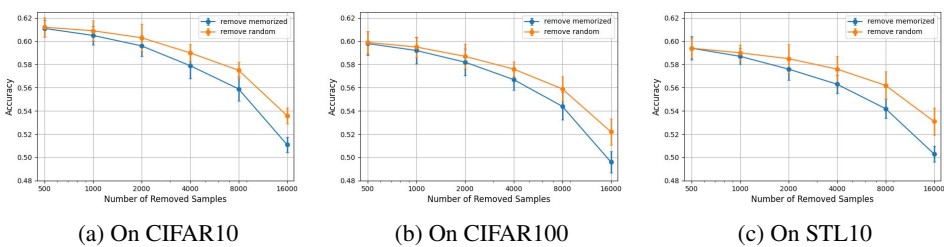

| (a) On CIFAR10 | (b) On CIFAR100 | (c) On STL10 |

Figure 7: **The influence of memorization on downstream generalization (CIFAR100).** We train an MAE model based on the VIT-tiny architecture on CIFAR100 and remove [500, 1k, 2k, 4k, 8k, 16k] most memorized vs. random data points from the encoder's training data. We measure downstream accuracy through linear probing on CIFAR10, CIFAR100, and STL10. The removal of memorized data points harms accuracy over all downstream tasks more than the removal of random data points.

which is a direct consequence of the high number of training iterations over the training data points. Encoders trained with SSL for the same number of epochs experience a significantly lower average memorization. When considering models that are aligned in downstream performance (supervised 10 epochs, vs. SSL 300 epochs), the computed memorization scores are comparable. Yet, as we will show in Appendix B.4 section, the two learning paradigms differ significantly in what types of data points they memorize.

**Comparing most memorized points from `SSLMem` and supervised learning.** We also assessed whether our score highlights the same data points as highly memorized as the metric proposed for supervised learning by (Feldman, 2020). Therefore, we trained a model $f$ and a model $g$ (both ResNet50 on CIFAR10) in a supervised manner. For best comparability with our results, we chose 40k data points overlap between the two models and 5k difference. On the 5k data points used to train $f$ but not $g$, we calculated the difference in softmax outputs between $f$ and $g$. The data points with the highest difference are the ones with the highest memorization according to Feldman (2020). To calculate our metric on the same models, we removed the classification layer and then calculated our metric on the output representations. We present the ten most memorized data points identified by both methods in Figure 8. Additionally, we analyze the overlap between both methods in Table 12. The results indicate that there is a roughly 50% overlap between the most memorized samples identified by both methods, and that the ranking between the samples is similarly consistent as the rankings by our `SSLMem` method over different SSL frameworks (see Table 13).

### B.4    ANALYSIS OF MEMORIZED SAMPLES

We set out to perform an in-depth analysis of the memorized samples. In particular, we compare the samples memorized by different SSL frameworks and architectures, and the difference in mem-

Table 11: **Impact of training paradigm on memorization.** We train a ResNet50 in a supervised manner with CIFAR10, and then remove the classification head to keep only the encoder part. We also train the ResNet50 with DINO in an SSL manner. We compare the average memorization and the linear probing accuracy of both resulting encoders.

| Training Method | Avg. Mem. | Linear Probing Acc. (%) |
|---|---|---|
| Supervised (100 epoch) | $0.398 \pm 0.010$ | $90.10\% \pm 1.34\%$ |
| SSL (100 epoch) | $0.314 \pm 0.011$ | $69.12\% \pm 0.87\%$ |
| SSL (300 epoch) | $0.327 \pm 0.009$ | $75.39\% \pm 1.15\%$ |
| Supervised (10 epoch) | $0.327 \pm 0.014$ | $75.16\% \pm 0.99\%$ |

Figure 8: **Most memorized data points identified with our `SSLMem` vs (Feldman, 2020).** We plot the ten data points from CIFAR10 with the highest memorization according to our memorization score and the metric for memorization in supervised learning proposed by Feldman (2020). We observe a high overlap between the most memorized samples identified by both methods.

orized samples between SSL and supervised learning. To obtain the highly memorized samples from supervised learning according to our score, we rely on the process described in the previous Section B.3.

We first visually inspect the samples that experience the highest memorization over all frameworks in Figure 9. Overall, the highest memorized samples seem to be more consistent between the different SSL frameworks than between SSL and supervised learning. This holds for both the supervised models, the one that is trained 100 epochs and the one that is trained 10 epochs, and thereby matches downstream performance of the SSL encoders (see Table 11). To quantify this visual impression, we analyze the rankings of memorization scores over the 5000 candidates for all different setups with a pairwise Kendall's Tau test. We present the results in Table 13. The null-hypothesis of the test is an absence of association between the two rankings, which means that when we have a p-value below $0.05$, *i.e.,* when we can reject the null-hypothesis, there is an association in the ranking. In the table, we indeed observe that the consistency between the rankings of memorization scores between different SSL frameworks is higher than the consistency between SSL and supervised models. In addition, among different SSL frameworks, the ones that share the same architecture (or training method) have a higher consistency.

### B.5 EXTENDED ANALYSIS ON MORE DATASETS AND MODEL ARCHITECTURES

We present an empirical evaluation on the effect of removing memorized vs. random samples on more fine-grained datasets in Table 14.

Additionally, in Table 15, we also report results for the same architectural family (ResNet) with different depths (ResNet50 vs ResNet30), and widths (Wide-ResNet). Our results show how the memorization score differs for various number of parameters and their arrangement and how it can influence the memorization score. We observe that with more parameters, encoders have higher memorization capacity.

### C MEMORIZATION SCORES

We make the following observations based on Figure 2c, Table 2, and Table 16:

Table 12: **Consistency between most memorized data points identified with our `SSLMem` vs (Feldman, 2020).** We depict the consistency between the [10,20,50,75,100,150,200] most memorized data points identified by our metric and the metric for supervised learning proposed by Feldman (2020). The first row shows the percentage of overlap and the second one the results of the statistical Kendall's Tau Test as $\tau$-Statistic / p-value.

| Within first X samples | 10 | 20 | 50 | 75 | 100 | 150 | 200 |
|---|---|---|---|---|---|---|---|
| % Overlap | 50.0% | 35.0% | 48.0% | 42.0% | 39.0% | 41.3% | 44.0% |
| Kendall's Tau Test | 0.099 / 0.584 | 0.124 / 0.332 | 0.162 / 0.107 | 0.158 / 5.42e-2 | 0.188 / 3.21e-2 | 0.174 / 9.66e-3 | 0.192 / 5.48e-4 |

Table 13: **Results of Kendall's Tau Test.** We test consistency of the rankings statistically of 5000 candidates over all models used for evaluation in this paper. Note that the score is symmetric. We repeat the values in gray for the reader's convenience.

| | MAE ViT-tiny | DINO ViT-tiny | DINO ResNet | SimCLR ResNet | Supervised ResNet, 100 epochs | Supervised ResNet, 10 epochs |
|---|---|---|---|---|---|---|
| MAE (ViT-tiny) | 1.0 / 0 | 0.235 / 2.2e-9 | 0.218 / 8.7e-9 | 0.207 / 5.1e-8 | 0.083 / 5.6e-5 | 0.104 / 1.3e-5 |
| DINO (ViT-tiny) | 0.235 / 2.2e-9 | 1.0 / 0 | 0.258 / 9.8e-12 | 0.214 / 1.0e-8 | 0.074 / 9.8e-4 | 0.092 / 3.2e-5 |
| DINO (ResNet) | 0.218 / 8.7e-9 | 0.258 / 9.8e-12 | 1.0 / 0 | 0.255 / 5.9e-11 | 0.091 / 3.4e-5 | 0.112 / 9.7e-6 |
| SimCLR (ResNet) | 0.207 / 5.1e-8 | 0.214 / 1.0e-8 | 0.255 / 5.9e-11 | 1.0 / 0 | 0.104 / 1.2e-5 | 0.096 / 2.2e-5 |
| Supervised (ResNet), 100 epochs | 0.083 / 5.6e-5 | 0.074 / 9.8e-4 | 0.091 / 3.4e-5 | 0.104 / 1.2e-5 | 1.0 / 0 | 0.131 / 2.3e-6 |
| Supervised (ResNet), 10 epochs | 0.104 / 1.3e-5 | 0.092 / 3.2e-5 | 0.112 / 9.7e-6 | 0.096 / 2.2e-5 | 0.131 / 2.3e-6 | 1.0 / 0 |

[1] the format for all data is $\tau$-Statistic / p-value

1. If the shared set $S_S$ is used in both $f$ and $g$ encoders, then the distribution of memorization scores for the data points from $S_S$ approximately follow Gaussian distribution with 0-mean. The memorization scores for the data points from $S_S$ are close to (concentrates at) 0.

2. The memorization scores for the candidates $S_C$ used only in the training of encoder $f$ are significantly above 0.

3. The memorization scores for the independent $S_I$ data points included in the training set of only $g$ are significantly below 0.

4. Data points from $S_C$ have statistically significantly and meaningfully higher memorization scores than those from $S_S$ and $S_E$.

5. Data points from $S_I$ have statistically significantly and meaningfully lower memorization scores than those from $S_S$ and $S_E$.

6. Memorization scores are close to 0 for both $S_E$ and $S_S$ and they approximately follow the Gaussian distribution. There is a difference in the mean scores between $S_E$ than $S_S$ since data points from $S_E$ are seen during training of neither $f$ nor $g$ while data points from $S_S$ are used for the training of both $f$ and $g$.

## D    EXTENDED ANALYSIS OF MEMORIZATION IN SSL

### D.1    ADDITIONAL BACKGROUND ON SSL

Many theoretical works on SSL, perform their analyses under the assumption that the training dataset $S$ in SSL comes from an underlying unlabeled data distribution $\mathcal{D}$ which is modeled as having $K$ disjoint latent classes $\Gamma_1, \ldots, \Gamma_K$ (Arora et al., 2019). Owing to the unlabeled nature of $S$, information about $\mathcal{D}$ and the latent classes is not known during training. However, the concept of latent classes helps define a structure of the data distribution and is helpful for analyzing the performance of $f$, *e.g.,* on downstream tasks. A commonly used assumption is that augmentations preserve the latent classes, *i.e.,* if $x \in \Gamma_k$, $\text{Aug}(x) \subseteq \Gamma_k$ (Wang et al., 2022).

### D.2    ANALYSIS OF SSL FRAMEWORKS AND ALIGNMENT

Standard SSL loss functions like the InfoNCE loss (see Equation (5)) can be decomposed into alignment and uniformity terms.

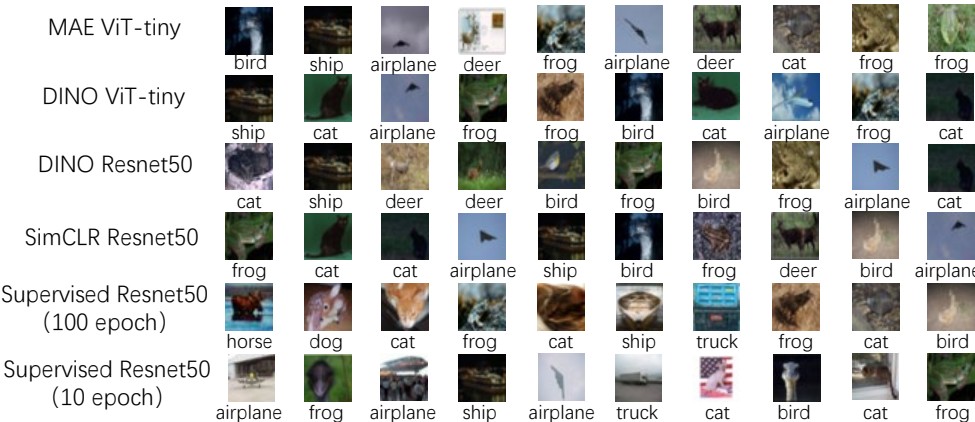

Figure 9: **Samples with highest memorization over different SSL frameworks and encoder architectures.** We depict per setup the top 10 data points with the highest memorization scores and their ground-truth labels.

Table 14: **Evaluating the effect of memorization on downstream tasks.** We pre-train a ViT-base with MAE on ImageNet and remove the top [10k, 20k] memorized vs. random data points. We easure downstream accuracy through linear probing on ImageNet, Food-101, and Flower102. The removal of memorized data points harms accuracy over all downstream tasks more than the removal of random data points.

| | Without removing | Removing 10000 | | Removing 20000 | |
| | | Memorized | Random | Memorized | Random |
|---|---|---|---|---|---|
| ImageNet | 68.21% ± 0.98% | 64.33%± 0.84% | 66.82%± 0.77% | 58.90± 1.05% | 62.88%± 0.77% |
| Food-101 | 58.96% ± 1.33% | 55.15%± 0.96% | 56.83%± 1.13% | 50.07%± 0.76% | 53.14%± 1.21% |
| Flower102 | 60.11% ± 1.12% | 56.27%± 1.14% | 58.08%± 0.89% | 51.48± 1.01% | 55.29± 0.93% |

$$\mathcal{L}(f, x) = - \underbrace{\mathbb{E}_{x^+ \sim Aug(x)} f(x)^T f(x^+)}_{\text{alignment}} + \underbrace{\mathbb{E}_{x^+ \sim Aug(x), \{x_i^-\}_{i=1}^l \sim S} \log \left( \exp(f(x)^T f(x^+)) + \sum_{i=1}^{l} \exp(f(x)^T f(x_i^-)) \right)}_{\text{uniformity}} \quad (5)$$

Zhang et al. (2022) show that MAE also implicitly aligns the mask-induced positive pairs. This is done through the masking, where the autoencoder is forced to reconstruct the same original image from two potentially disjoint (differently masked) views. MAE aligns explicitly in the output space, however, the decoder part is very shallow ($< 10\%$ of the encoder) and translates to the alignment in the latent feature space. This directly applies to other SSL frameworks which also append the additional shallow projection heads to the encoders and explicitly align only the final outputs instead of representations. The main difference between MAE and other SSL frameworks is a lack of the uniformity in the representation space, where the learned features lie in a low dimensional subspace (Hua et al., 2021; Jing et al., 2022). The recovery of uniformity in MAE requires further enhancement of its loss with the additional term $\mathbb{E}_x \mathbb{E}_{x^-} (f(x)^T f(x^-))^2$, where $x^-$ is a negative pair of $x$ (different data points than $x$). This reduces our definition of memorization to alignment (with augmentations) as the common property of the representations across all the considered SSL methods.

### D.3 INTUITION BEHIND OUR MEMORIZATION SCORE

To provide intuition behind why it is meaningful to define memorization based on the alignment loss of data points and to use the leave-one-out style definition, we present a simple example with a one-dimensional input and latent space (so that the data can be defined with the $x$ coordinate and the representation with the $y$ coordinate) which we visualize in Figure 10. The example highlights how a

Table 15: **Evaluation of memorization on different architectures.** We train encoders with different backbone architectures using SimCLR on CIFAR10. We report the average memorization of `SSLMem` together with the resulting linear probing accuracy and the number of model parameters.

| Architecture | `SSLMem` | Acc. | # of Parameters |
|---|---|---|---|
| Wide-ResNet50-2 | $0.350 \pm 0.008$ | $81.23\% \pm 1.01\%$ | 69M |
| ResNet50 | $0.339 \pm 0.011$ | $77.12\% \pm 1.42\%$ | 25M |
| ResNet18 | $0.315 \pm 0.013$ | $71.07\% \pm 1.08\%$ | 11M |

Table 16: **More results of statistical t-tests.** We provide evidence that the memorization scores are: for $S_C$ significantly above 0, for $S_I$ significantly below 0. The test is inconclusive for the hypothesis that scores for $S_S$ are equal or below 0. This indicates that the scores are not significantly different from 0. We also observe that the scores for $S_E$ are significantly above 0 but still significantly below the scores for $S_C$.

| Null Hypothesis | p-value | effect size |
|---|---|---|
| $\mathcal{H}_0 := m(S_C) \leq 0$ | 0 | 101.24 |
| $\mathcal{H}_0 := 0 \leq m(S_I)$ | 0 | 100.42 |
| $\mathcal{H}_0 := 0 = m(S_S)$ | 0.471 | 0.821 |
| $\mathcal{H}_0 := m(S_E) \leq 0$ | 0 | 53.23 |
| $\mathcal{H}_0 := m(S_C) \leq m(S_E)$ | 0 | 60.44 |

data point $x$ selected either as a *standard* in-distribution or *outlier* data point impacts the training algorithm and can cause different levels of memorization.

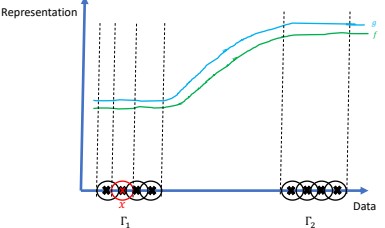
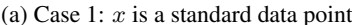
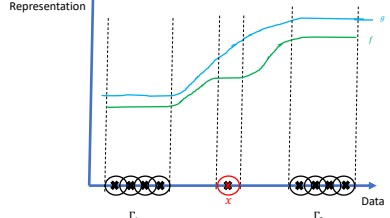

(a) Case 1: $x$ is a standard data point.    (b) Case 2: $x$ is an outlier data point.

Figure 10: **Intuition for the memorization of data points.** We provide a simple one-dimensional example to build the intuition behind our definition of memorization. On the x-axis, we depict the input dimension and on the y-axis the representations returned by encoders $f$ and $g$. In (a), the data point $x$ which $f$ is trained on and $g$ is not, is a standard "inlier" data point. In (b), $x$ is an atypical data point or "outlier".

Assume without the loss of generality that there are two latent classes, $\Gamma_1, \Gamma_2$ in the data space. The augmentation sets form regions around the training data points which are represented by circles around the points in Figure 10. We assume that the latent classes have central clusters where all data points have augmentation sets which overlap with at least one other augmentation set (this is similar to (Huang et al., 2023)). Let these central clusters be $\Gamma_1^0 \subseteq \Gamma_1$ and $\Gamma_2^0 \subseteq \Gamma_2$ respectively. To further simplify the example, we will assume that the overlap is such that for any $x_i$ in $\Gamma_1^0$ or $\Gamma_2^0$, the whole augmentation set $Aug(x_i)$ is involved in the overlap.

In our example, we consider the effect of training encoders $f \in \mathcal{F}$, *i.e.*, encoders trained on $S$ including $x$, and encoders $g \in \mathcal{G}$, *i.e.*, encoders trained on $S \setminus x$. We assume that encoders $f$ and $g$ are trained until their respective training losses (over all training datapoints individually) are smaller than some constant $c$. Specifically, we assume a stronger notion of alignment, namely $c$-strong alignment:

**Definition 1** ($c$-Strong Alignment). *We say that an encoder $f$ satisfies $c$-strong alignment on datapoint $x_i$ if $\forall\, x', x'' \in Aug(x_i), d(f(x'), f(x'')) \leq c$.*

The difference between this definition and the standard definition of alignment is that the expected value has been replaced with a for all operator. We also assume that all possible functions $f$ and $g$ are $L$-Lipschitz continuous *i.e.*, $d(f(x_1), f(x_2)) \leq L||x_1 - x_2|| \; \forall x_1, x_2$. This assumption has been

used by prior works on SSL e.g. (Huang et al., 2023). Finally, when dealing with representations, we assume that they are normalized with $||f(x_i)||_2 = ||g(x_i)||_2 = r$.

To analyze memorization with our leave-one-out definition (Definition 2), we will examine two main cases. First, we consider the case where $x$ is a standard data point from $\Gamma_1^0$, as in Figure 10a. Then, we know that every point in $\text{Aug}(x)$ is also a member of $\text{Aug}(x_i)$ for some $i$. Thus, even though there is no explicit constraint on the alignment of $g$ during training, this overlap will mean implicitly that $d(g(x'), g(x'')) \leq b \cdot c$ for any $x', x'' \in \text{Aug}(x)$ so that $\mathcal{L}_{\text{align}}(g, x) \leq b \cdot c$ for all $g \in \mathcal{G}$. Here, $b$ is a constant and is related to the fact that there may be multiple augmentation sets that need to be traversed, when going from $x'$ to $x''$. Meanwhile, by assumption during training, we know directly that $\mathcal{L}_{\text{align}}(f, x) \leq c \ \forall f \in \mathcal{F}$.

Second, we consider the case where data point $x$ is an outlier as in Figure 10b. In this case, the augmentation set $\text{Aug}(x)$ does not overlap with other data points from $S$. Then there is no explicit or implicit constraint in the training objective for encoders $g$ and thus no upper bound on the alignment of $g$ on $x$. Therefore, the overall function class $\mathcal{G}$ consisting of all possible encoders $g$ is now a superset of $\mathcal{G}$ from the first example. Hence, the alignment of $g$ on $x$ now has a strictly higher value than in our first example. Meanwhile, encoders $f$ have the same constraint so that $\mathcal{F}$ will have the same alignment loss as in the first case. Therefore, considering the difference $\mathbb{E}_{g \in \mathcal{G}} \mathcal{L}_{\text{align}}(g, x) - \mathbb{E}_{f \in \mathcal{F}} \mathcal{L}_{\text{align}}(f, x)$, this case has a strictly higher difference and thus higher memorization scores.

To summarize, in the first case the behaviour of models $f \in \mathcal{F}$ and $g \in \mathcal{G}$ does not differ significantly on $x$ due to the implicit constraints. In contrast, in the second case, there is a more significant difference where only model $f$'s behavior is significantly shaped by $x$, indicating a higher level of memorization.

## D.4 THE LINK BETWEEN MEMORIZATION AND GENERALIZATION

In the context of supervised learning, Feldman (2020) has shown that memorization of outlier data points is required to achieve a close to optimal generalization error on natural data distributions, where data often follows a long-tailed distribution. Even though the concept of labels does not exist in SSL, we show in this section that memorization of outlier examples is still highly relevant to obtaining a good generalization. While we measure memorization on the level of SSL encoders' representations, following prior work, *e.g.,* Huang et al. (2023); Cabannes et al. (2023), we focus our notion of generalization on the level of *downstream tasks*, as these types of tasks are typical use-cases of SSL models. To this end, in this section, we consider classification downstream tasks, and, as discussed in the problem setup, we consider the error that classifier $G_f$ achieves on downstream tasks from the same/different distributions as the unlabeled encoder training data. For analysis purposes, we assume that $G_f$ is a nearest centroids classifier so that $G_f(x) = \arg\min_{k \in [K]} ||f(x) - \mu_k||$, with $\mu_k = \mathbb{E}_{x \in \Gamma_k} f(x)$. Huang et al. (2023) has shown that this is a special case of a general linear classifier.

When revisiting the intuition on memorization described in the previous section (Appendix D.3), we observe that for encoders $g \in \mathcal{G}$, the alignment loss is unlikely to be low in regions of the data space with outliers. In contrast, for encoders $f \in \mathcal{F}$, we observe good alignment over regions where outliers are present. We visualize this effect for a synthetic two dimensional data distribution in Figure 11.

We now describe this property more concretely and provide guarantees for the associated error that the downstream classifier will achieve. Our analysis will be centered around a particular outlier datapoint $x$ and the generalization error will be estimated with a testing dataset $S_{\text{test}} = \{z_1, \ldots, z_l\}$, consisting of points not used during training. We start by presenting some supporting definitions which will be helpful for this analysis.

**Definition 2** ($\sigma$-overlap)**.** *We say that the augmentation set of a datapoint $z$ satisfies $\sigma$-overlap if there exists a region $Aug^0(z) \subseteq Aug(z)$ which overlaps with the augmentation set of a training datapoint $x_i \in S$ and so that $P[b \in Aug^0(x)] \geq \sigma \cdot P[b \in Aug(x)]$.*

**Definition 3** ($\beta$-close)**.** *We say that a datapoint $z$ is $\beta$-close to a training datapoint $x_i \in S$ if $\min_{x_i' \in Aug(x_i)} ||x_i' - z|| = \beta$.*

The following lemma presents a simple upper bound on the difference between the representations $f(x)$ and $f(z_i)$ for any test datapoint $z_i$.

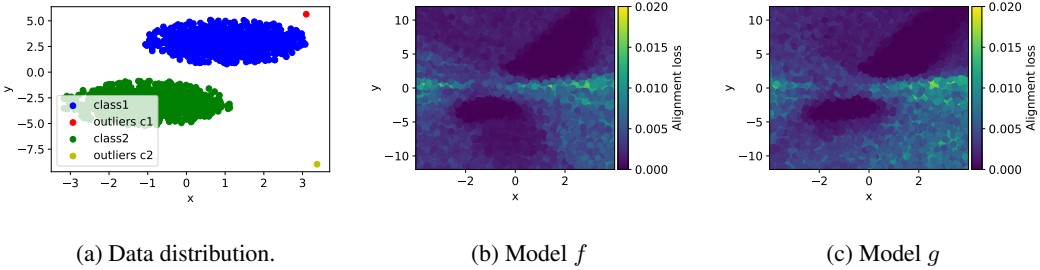

(a) Data distribution.  (b) Model $f$  (c) Model $g$

Figure 11: **Training with outliers yields lower alignment loss over additional regions of the representation space.** (a) We generate a two dimensional distribution of data that consists of two latent classes. For each latent class, we have a central part and one outlier example. Then, we train two encoders with the InfoNCE loss—$f$ with the outliers, and $g$ without—that map from the data to a one dimensional representation space. (b) shows that by training with outliers, the resulting model gets a lower alignment loss in locations where there were outliers, whereas we see in (c) that when training without outliers, only the alignment loss in the main data cluster regions is decreased.

**Lemma 4.** *Given an encoder $f$ satisfying $c$-strong alignment over point $x$ and a test datapoint $z_i \in S_{test}$ which satisfies $\beta_i$-closeness to point $x$, $d(f(x), f(z_i)) \leq L\beta_i + c$*

*Proof.* Follows directly from the triangle inequality. We have $f(x) - f(z_i) = f(x) - f(x') + f(x') - f(z_i)$ where $x'$ is the point obtained from Definition 3. Then $d(f(x), f(z_i)) = ||f(x) - f(z_i)||_2 \leq ||f(x) - f(x')||_2 + ||f(x') - f(z_i)|| \leq c + L \cdot \beta_i$ where $c$-strong alignment and the Lipschitz property have been used. $\square$

Similarly, we can upper bound the alignment loss $\mathcal{L}_{\text{align}}(f, z_i)$ with the following lemma.

**Lemma 5.** *Given an encoder $f$ with $\mathcal{L}_{align}(f, x) \leq c$ and point $z_i$ which satisfies $\sigma$-overlap with point $x$, $\mathcal{L}_{align}(f, z_i) \leq \sigma \cdot c + (1 - \sigma) \cdot L \cdot \mathbb{E}_{z', z'' \in (Aug(z_i) \setminus Aug(z_i) \cap Aug(x))}||z' - z''||$*

*Proof.*

$\mathcal{L}_{\text{align}}(f, z_i) = \mathbb{E}_{z', z'' \in \text{Aug}(z_i)} d(f(z'), f(z''))$

$= P[b \in \text{Aug}(x) \cap \text{Aug}(z_i) | b \in \text{Aug}(z_i)] \cdot \mathbb{E}_{z', z'' \in \text{Aug}(x) \cap \text{Aug}(z_i)} d(f(z'), f(z''))$

$+ P[b \in \text{Aug}(z_i) \setminus (\text{Aug}(x) \cap \text{Aug}(z_i)) | b \in \text{Aug}(z_i)] \cdot \mathbb{E}_{z', z'' \in \text{Aug}(z_i) \setminus (\text{Aug}(x) \cap \text{Aug}(z_i))} d(f(z'), f(z''))$

$\overset{(a)}{\leq} \sigma \cdot c + (1 - \sigma) \cdot \mathbb{E}_{z', z'' \in \text{Aug}(z_i) \setminus (\text{Aug}(x) \cap \text{Aug}(z_i))} d(f(z'), f(z''))$

$\overset{(b)}{\leq} \sigma \cdot c + (1 - \sigma) \cdot L \cdot \mathbb{E}_{z', z'' \in \text{Aug}(z_i) \setminus (\text{Aug}(x) \cap \text{Aug}(z_i))} ||z' - z''||$

where (a) follows from the definition of $\mathcal{L}_{\text{align}}(f, x)$ and (b) follows from Lipschitzness. $\square$

Note that for the purpose of these lemmas, we assume that $d$ is the $\ell_2$ norm. We are now ready to analyze the relationship between the generalization error and memorization. We will compare between two algorithms $\mathcal{A}_1, \mathcal{A}_2$ where $\mathcal{A}_1$ has a greater degree of memorization on data point $x$. We will then show that models $f_1 \sim \mathcal{F}_1 = \mathcal{A}_1(S)$ will likely have a lower generalization error than models $f_2 \sim \mathcal{F}_2 = \mathcal{A}_2(S)$. We start by selecting the test points $z_i$ which are $\beta_i$ close to $x$ for $\beta_i \leq \beta$, where $\beta$ is a selected upper bound, *e.g.,* $\beta = \frac{c}{L}$. Without loss of generality, let these points be $z_1, \ldots, z_t$. Given that $x$ is an outlier datapoint, we now treat it as being part of a $K + 1$st latent class, where the only datapoint from this latent class to appear in the training dataset $S$ is $x$. In other words, $x$ can be seen as a singleton example (Feldman, 2020). On the basis of closeness in the data space, we will then also assume that all of $z_1, \ldots, z_t$ belong to the same latent class as $x$ *i.e.,* to $\Gamma_{K+1}$.

We now claim that for cases where the complexity of encoders learnt by algorithms $\mathcal{A}_1, \mathcal{A}_2$ is the same and where learning a good representation on $x$ is not trivial, $\mathbb{E}_{f_1 \sim \mathcal{F}_1} \mathcal{L}_{\text{align}}(f_1, x) <$

Table 17: **Training on the most memorized data points**. We traine a ResNet50 on SimCLR with CIFAR10 (25k training data points) and calculated the memorization score over all data points. We then train again from scratch with the [25k (all), 24k, 22k, 20k, 16k, 12k] most memorized data points. We report linear probing accuracy on CIFAR10, CIFAR100, and STL10. Our results highlight that by training on the most memorized data points, we can outperform or match the performance of the encoder trained on the full 25k data points.

| Retained Points | CIFAR10 | CIFAR100 | STL10 |
|---|---|---|---|
| 25k (full encoder) | $63.3\% \pm 0.92\%$ | $61.1\%\pm1.14\%$ | $61.6\%\pm0.83\%$ |
| 24k (most memorized) | **$64.4\% \pm 1.03\%$** | **$61.3\%\pm0.98\%$** | **$61.7\pm1.18\%$** |
| 22k (most memorized) | **$63.8\% \pm 0.76\%$** | **$61.8\pm1.24\%$** | **$62.4\pm1.05\%$** |
| 20k (most memorized) | $63.2\% \pm 1.07\%$ | $60.8\%\pm0.68\%$ | $61.1\pm1.05\%$ |
| 16k (most memorized) | $61.8 \pm 1.11\%$ | $58.4\%\pm0.91\%$ | $59.9\pm0.89\%$ |
| 12k (most memorized) | $59.7\% \pm 0.74\ \%$ | $55.6\%\pm1.32\%$ | $55.2\pm1.24\%$ |

$\mathbb{E}_{f_2\sim\mathcal{F}_2}\mathcal{L}_{\text{align}}(f_2,x)$. This is because for models $f_2$, the point $x$ is not memorized and thus $\mathcal{L}_{\text{align}}(f_2,x) \approx \mathcal{L}_{\text{align}}(g_2,x)$ which we can expect will be larger than the alignment loss when including $x$ as a training data point. Note that here we also use the fact that the range of possible values the alignment loss can take are the same for both algorithms since $0 \leq \mathcal{L}_{\text{align}} \leq 2r$ as a result of the representations being normalized. With this claim, we can then assume that encoders $f_1$ satisfy $c$-strong alignment for some value of $c$, based on which Lemma 4 will imply that $d(f_1(x), f_1(z_i)) \leq L\beta + c$ for $1 \leq i \leq t$. Meanwhile, encoders $f_2$ do not have such a guarantee and thus while there may exist some encoders $f_2$ which do satisfy this bound, in expectation we will likely have $d(f_2(x), f_2(z_i)) > d(f_1(x), f_1(z_i))$.

We now analyze the error of the linear classifier on the datapoints $z_1, \ldots, z_t$. From the form of the models $G_f$, we know that the decision rule is to select class $K + 1$ if $||f(z_i) - \mu_{K+1}|| \leq ||f(z_i) - \mu_k|| \ \forall k \leq K$. In this case, since $x$ is the only training point from latent class $K + 1$, $\mu_{K+1} = f(x)$. Now while reasoning about the class centers is difficult based on a single datapoint changing and the different algorithms that are used, we note that a smaller value of $d(f_1(x), f_1(z_i))$ can help encoders $f_1$. Given that the upper bound on the alignment (Lemma 4) is certain to hold for encoders $f_1$, we can thus have provable guarantees on the error the classifier achieves over these testing datapoints (assuming sufficient class center separation). For encoders $f_2$, it is unlikely that all encoders will lead to good predictive accuracy of the classifiers. Therefore, we can see a relationship between memorization of the datapoint $x$ and the (average) error of the classifiers on nearby datapoints (which is a component of the overall generalization error). This can thus show a potential correlation between memorization and generalization error. We leave a more thorough investigation of this concept to future work.

## D.5 PRACTICAL IMPLICATIONS OF MEMORIZATION

While our work is interested in studying the fundamental properties of SSL memorization to deepen our understanding of this learning paradigm and to reveal similarities and dissimilarties with supervised learning and between different SSL frameworks, it also has some practical implications.

**Data Privacy.** Our method supports studying which data points experience highest memorization by the encoder. These data points are particularly prone to privacy leakage. Based on the insights from our memorization score, depending on the type of use case of such encoders, appropriate action (such as differential privacy, potentially with stronger guarantees for the memorized data points (Jorgensen et al., 2015)) can be taken during or after the training to limit the leakage.

**Coreset Selection.** We also show that our method is related to the research line of coreset selection (Paul et al., 2021; Sener & Savarese, 2018; Tsang et al., 2005), *i.e.,* the identification of (smaller) data subsets that can be leveraged for training more efficiently while obtaining the same performance. In the same setup as Figure 3 in Section 4.4 4.4, we trained a ResNet50 on SimCLR with CIFAR10 (25k training data points) and calculated the memorization score over all data points. We then trained the model again from scratch with the [25k (all), 24k, 22k, 20k, 16k, 12k] most memorized data

points. We report the downstream accuracy on CIFAR10, CIFAR100, and STL10 in Table 17. Our results highlight that by training only on the subset of most memorized data points, we can even outperform the encoder trained on the full dataset, or match its performance with a significantly smaller training dataset (up to 25% smaller). Thereby, our method can lead to new learning strategies that could dramatically (1) reduce training times and (2) reduce data and memory requirements for the SSL encoders (which are both extremely high under current methods).

