# OpenReview forum: "Memorization in Self-Supervised Learning Improves Downstream Generalization"
_ICLR.cc/2024/Conference — ICLR 2024 poster_

### Official Review · Reviewer_ukN9 · 2023-10-13

**Soundness:** 2 fair
**Presentation:** 3 good
**Contribution:** 2 fair
**Rating:** 6
**Confidence:** 4

**Summary:**

This paper proposes a novel framework that defines memorization within the context of self-supervised learning (SSL). This definition compares the difference in alignment of representations for data points and their augmented views returned by encoders that were trained on these data points and encoders that were not. Empirical analysis on diverse encoder architectures and data sets demonstrate that significant fractions of training data points experience high memorization in SSL, and memorization is essential for encoders to achieve higher generalization performance on different downstream tasks.

**Strengths:**

- The authors propose a novel definition that generalizes memorization effect to self-supervised learning, which can be useful for understanding model generalization
- This paper is easy to understand

**Weaknesses:**

- The proposed framework lacks enough practical implications. It is not so clear how we should treat samples with higher/lower memorization scores differently.
- Several claims are subject to challenge, as can be found in Questions part.
- Empirical results may be further improved, including more different architectures and data sets.

**Questions:**

- The definition of (per-example) memorization score is a bit strange. From my perspective, a higher memorization score indicates that sample $x$ has larger impact to model performance. As such, it seems natural that pre-trained models with higher memorization perform well on downstream tasks: if not, it means all training samples do not contribute much to pre-training, and it will be strange that pre-training on such data set produces good models. Some more explanations may be needed on why we need to introduce memorization score here, instead of some other performance metrics.
- Also, it would be better if the authors can show some connections between other metrics on generalization (examples may be found in [1]) and the proposed memorization score. It would be useful to see how good final performance comes from higher memorization scores beyond simply putting them together.
- While the authors claimed that their experiments cover different architectures and downstream data sets, the experiments seem a bit restricted from my perspective. For architectures, it would be better if the authors can report i) same architecture (e.g., ResNet) with different depths, widths, etc. and ii) similar number of parameters with different architectures (ViT and ResNet here). That will make the experimental results more comprehensive and we may also gain some insights on how memorization differs across different architectural settings
- Also, downstream data sets used in current paper (CIFAR-10/100, SVHN, STL-10, ImageNet) are all coarse-grained general classification data sets. The authors may consider also adding some experiments on fine-grained data sets (e.g, Food-101 or Flower102, which are popularly used in literature) and see if their conclusions are changed.
- Regarding experiments on removing samples for downstream tasks, the performance gap seems not so large. What will happen if we directly try to remove some samples to obtain largest performance drop? Will these samples found have larger memorization scores? Some additional experiments are welcome.
- I also wonder how it is connected to research on coresets [2], which aims to find a (small) subset of training data that can help obtain models with performances close to full training data. The authors may report something opposite to Figure 3 and Table 3: solely train on samples with high memorization scores, and see how the model works.

References:

[1] Fantastic Generalization Measures and Where to Find Them. ICLR 2020

[2] Deep Learning on a Data Diet: Finding Important Examples Early in Training. NeurIPS 2021

---

> ### Author Response · Authors · 2023-11-18
> **Thank you for the review, Practical Implications, Further Experimentation**
>
> We thank the reviewer for the detailed comments and are glad that the reviewer recognizes our extensive “empirical analysis on diverse encoder architectures and data sets” and finds our framework “useful for understanding model generalization”. Below we address the points and questions raised by the reviewer one by one.
>
> >_**The proposed framework lacks enough practical implications. It is not so clear how we should treat samples with higher/lower memorization scores differently.**_
>
> As pointed out by the reviewer, our work is “useful for understanding model generalization”. We study the memorization and generalization properties of self-supervised learning (SSL) to deepen our understanding of this new learning paradigm and to reveal similarities and differences within different SSL frameworks and with respect to the well-studied supervised learning.
>
> **Coreset Selection:** Following the reviewer’s suggestion, we ran additional experiments to investigate the suitability of our method for coreset selection (see our last answer). Our results highlight that by training only on the subset of most memorized data points, we can even outperform the encoder trained on the full dataset, or match its performance with a significantly smaller training dataset (up to 25% smaller). Thereby, our method can lead to new learning strategies that could dramatically reduce (1) training and computation time and (2) data and memory requirements for the SSL encoders (which are both extremely high under current methods).
>
> **Data Privacy:** Our method supports studying which data points experience highest memorization by the encoder. These data points are particularly prone to privacy leakage. Based on the insights from our memorization score, depending on the type of use case of such encoders, appropriate actions (such as differential privacy, potentially with stronger guarantees for the memorized data points [A, B]) can be taken during or after the training to limit the leakage.
>
> We added a new Appendix Section D.5 to the updated version of our paper to highlight these practical implications of our framework.
>
> [A] Alaggan, Mohammad, Sébastien Gambs, and Anne-Marie Kermarrec. "Heterogeneous Differential Privacy." Journal of Privacy and Confidentiality 7, no. 2 (2016).
> [B] Yaodong Yu, Maziar Sanjabi, Yi Ma, Kamalika Chaudhuri, Chuan Guo. “ViP: A Differentially Private Foundation Model for Computer Vision.” https://arxiv.org/abs/2306.08842 (June 2023).
>
>
> >_**Empirical results may be further improved, including more different architectures and data sets.**_
>
> Even though the reviewer points out in the summary that the work already carried out an "Empirical analysis on diverse encoder architectures and data sets" and Reviewer mj5F notes that “The authors have done an excellent job with their extensive empirical analysis.”, we were happy to further improve the empirical results following the recommendations and ran experiments with more fine-grained datasets (Food-101 and Flower102, as suggested by the reviewer), and different architectures (WideResNet and ResNet34, based on the reviewer’s advice). The results are included below with the concrete questions and in the updated version of the paper in Section B.5 in the Appendix.

---

> ### Author Response · Authors · 2023-11-18
> **Definition of Memorization, Connection to other Generalization Estimates**
>
> >_**The definition of (per-example) memorization score is a bit strange. From my perspective, a higher memorization score indicates that sample x has larger impact to model performance. As such, it seems natural that pre-trained models with higher memorization perform well on downstream tasks: if not, it means all training samples do not contribute much to pre-training, and it will be strange that pre-training on such data set produces good models. Some more explanations may be needed on why we need to introduce memorization score here, instead of some other performance metrics.**_
>
> Our work’s main focus is on measuring the impact of individual data points on the performance of an encoder. Therefore, it is necessary to compute a per-example memorization score. In particular, we are interested in understanding which data points that contribute most to the training algorithm (and experience the highest memorization). The relevance of this research area, and the interest to study a per-example impact on models and training is based on a long line of work in that area. For example, Arpit et al., [2017] study memorization in supervised learning through the lens of hardness to learn individual examples throughout training. Our definition is inspired by Feldman [2020] and Feldman&Zhang [2020] who study the per-example memorization in supervised learning with a leave-one-out definition and use it to identify the data points that experience highest memorization.
>
> Regarding the use of other performance metrics, we would like to point out that our work does not consider the limited setup studied in prior work where the training data is the same as the data used for evaluating performance. Instead, we pretrain an encoder on a dataset (e.g., CIFAR10 in Figure 3), and evaluate the performance of the resulting encoder on downstream tasks from entirely different data distributions (CIFAR100, STL10). In this setup, we cannot tie back easily the performance on the test data to the training data because performance of the test data only results from representations that are based on the training data (but cannot be traced back to individual training data points).
>
> >_**Also, it would be better if the authors can show some connections between other metrics on generalization (examples may be found in [1]) and the proposed memorization score.**_
>
> We would first like to point out that all the metrics proposed in [1] are defined w.r.t. (with respect to) to the model and its full training dataset (e.g., sharpness of optima, norm of weights, or speed of optimization algorithm). Our memorization score is defined w.r.t. individual data points and outputs a per-data point measure. This is fundamentally different from metrics in [1].
> Still, we can use the per-data point scores to perform experiments that consider the model as a whole, for example, as we do in Table 1 by reporting the average memorization score per encoder and over all points from the candidate set $S_C$. To follow the reviewer’s suggestion and compare with metrics from [1], we performed additional experiments and analyzed the connection between our memorization score on the MAE trained with ViT-tiny on CIFAR10 from Figure 3 with another metric, namely the parameter norm of the trained encoders. This is one of the few metrics that does not rely on a downstream task, and can, therefore, be evaluated in the same setup as our memorization score. We present the $\ell_2$ norm of the parameters and the linear probing test accuracy on CIFAR10 from Figure 3:
> |  | No removal    | 2k random     | 2k memorized  | 4k random     | 4k memorized  | 8k random     | 8k memorized  | 16k random    | 16k memorized |
> |---------------|---------|---------|----------|-----------|---------------|------------|---------------|---------------|---------------|
> | $\ell_2$ norm of params  | 103.58        | 104.62        | 105.81        | 105.90        | 107.34        | 110.07        | 114.83        | 122.43        | 128.05  |
> | Acc.  | 63.3% ± 0.92% | 62.3% ± 0.77% | 61.5% ± 0.85% | 61.1% ± 1.02% | 59.7% ± 1.06% | 59.5% ± 0.94% | 57.3% ± 1.19% | 55.6% ± 1.01% | 51.9% ± 1.30% |
>
> The fundamental approach in machine learning to obtain better generalization is by the $\ell_2$ regularization strategy, which drives the weights closer to the origin. Similarly, we observe in the above Table and Figure 3a (in the paper) that with lower $\ell_2$ norm of the parameters, the generalization of the encoder is higher (with no removal vs with the removal of data points). When we consider the removal of random vs memorized data points, we see the same trend, where the $\ell_2$ norm of the parameters is higher after removing memorized points, which causes a higher drop in generalization. This indicates that this metric could also be used as a proxy for generalization, however only on the full encoder level, and not on a per-data point level like our memorization score, which also allows, e.g., the coreset selection.

---

> ### Author Response · Authors · 2023-11-18
> **Connection between Memorization and Generalization, More Architectures, More Complex Datasets**
>
> >_**It would be useful to see how good final performance comes from higher memorization scores beyond simply putting them together.**_
>
> To provide reasoning why increased memorization leads to higher downstream performance, we would like to point the reviewer to Appendix D.4 “The Link between Memorization and Generalization” of our original submission. There, we provid the formal intuition on why memorized samples (from small sub-populations) of the encoder’s training data lead to improved generalization. Additionally, we experimentally show that the memorized samples cause a lower alignment loss in larger regions of the representation space. Figure 11 shows that without the inclusion of memorized samples (Figure 11c), the alignment loss is only low in the regions where the main data distribution clusters. However, with the memorized (long-tail, atypical) samples (Figure 11b), larger regions of the representation space exhibit a lower alignment loss. A low alignment loss means that similar inputs obtain similar representations, which helps a downstream classifier to learn better decision boundaries and classify samples in these regions better which boosts generalization.
>
> >_**While the authors claimed that their experiments cover different architectures and downstream data sets, the experiments seem a bit restricted from my perspective. For architectures, it would be better if the authors can report i) same architecture (e.g., ResNet) with different depths, widths, etc. and ii) similar number of parameters with different architectures (ViT and ResNet here). That will make the experimental results more comprehensive and we may also gain some insights on how memorization differs across different architectural settings.**_
>
> We also report results for the same architecture (e.g., ResNet) with different depths, and widths (Wide ResNet). It shows how the memorization score differs for various number of parameters and their arrangement and how it can influence the memorization score. We observe that with more parameters, models have higher memorization capacity. We added these new results to Appendix B.5 of the updated paper.
>
> |  | **Avg. Mem.** | **Acc.** | **# of Parameters** |
> |---|-----|-----|------|
> | SimCLR with Wide-ResNet50-2 | 0.350 ± 0.008 | 81.23 ± 1.01%  | 69M  |
> | SimCLR with ResNet50  | 0.339 ± 0.011 | 77.12% ± 1.42% | 25M   |
> | SimCLR with ResNet18  | 0.315± 0.013  | 71.07 ± 1.08%  | 11M   |
>
>
> >_**Also, downstream data sets used in current paper (CIFAR-10/100, SVHN, STL-10, ImageNet) are all coarse-grained general classification data sets. The authors may consider also adding some experiments on fine-grained data sets (e.g, Food-101 or Flower102, which are popularly used in literature) and see if their conclusions are changed.**_
>
> We also added more experiments on fine-grained downstream data sets, namely Food-101 and Flower102 to obtain results similar to Figure 3 and Table 3.
> We pre-trained a ViT-base encoder with MAE on ImageNet and then removed the 10k and 20k most memorized and random data points before training again. Then, we trained downstream classifiers on ImageNet, Food-101, and Flower102 on top of the resulting 5 encoders, and report below the downstream accuracies.
>
> | **Downstream Accuracy** | **Without Removing** | **Remove 10k** | **Remove 10k** | **Remove 20k** | **Remove 20k** |
> |-------|-----|----|----|---|-----|
> |   |   | **Memorized**  | **Random**     | **Memorized**  | **Random**     |
> | ImageNet  | 68.21% ± 0.98%       | 64.33%± 0.84%  | 66.82%± 0.77%  | 58.90± 1.05%   | 62.88%± 0.77%  |
> | Food-101 | 58.96% ± 1.33%       | 55.15± 0.96%   | 56.83± 1.13%   | 50.07± 0.76%   | 53.14± 1.21%   |
> | Flower102  | 60.11% ± 1.12%       | 56.27%± 1.14%  | 58.08%± 0.89%  | 51.48± 1.01%   | 55.29± 0.93%   |
>
> Our results show the same effect as the experiments from our original submission: also on these more fine-grained datasets, removing memorized data points harms downstream performance (on the same and different distributions) significantly more than removing random training points. We also added these new results to Appendix B.5 of the updated paper.

---

> ### Author Response · Authors · 2023-11-18
> **Performance Gap, Coresets**
>
> >_**Regarding experiments on removing samples for downstream tasks, the performance gap seems not so large. What will happen if we directly try to remove some samples to obtain the largest performance drop? Will these samples found have larger memorization scores? Some additional experiments are welcome.**_
>
> Our metric is able to identify exactly the data points whose removal causes the performance drop. Could the reviewer suggest a method to identify the samples that are not marked by our metric as highly memorized but can cause a higher performance drop than the highly memorized training data points? A naive approach would be to check each point individually by training with or without this data point, however, it is computationally too expensive to run. Our method provides a good proxy to find data points that have a high impact on the performance.
> Regarding the identification of samples that cause the largest performance drop, we would like to point out again that we consider pretraining with one dataset (e.g., CIFAR10 in Figure 3), and evaluating performance of the resulting encoder on entirely different data. The pretraining data hence only influences representations which makes it hard to quantify the impact that a pretraining data point has on the downstream performance.
>
> The new experiments on the more fine-grained datasets (see previous answer) highlight a much larger performance gap (7-8%), especially for the removal of 20k samples. This is a significant difference and enforces the findings from the original submission.
>
> >_**I also wonder how it is connected to research on coresets [2], which aims to find a (small) subset of training data that can help obtain models with performances close to full training data. The authors may report something opposite to Figure 3 and Table 3: solely train on samples with high memorization scores, and see how the model works.**_
>
> We thank the reviewer for this excellent question and performed the suggested experiments. In the same setup as Figure 3 in Section 4.4, we trained a ResNet50 on SimCLR with CIFAR10 (25k training data points) and calculated the memorization score over all data points. We then trained the model again from scratch with the [25k (all), 24k, 22k, 20k, 16k, 12k] most memorized data points. We report the downstream accuracy on CIFAR10, CIFAR100, and STL10  in the table below.
>
> We observe that indeed by training only on the most memorized data points, we can achieve higher downstream accuracy on the same and different downstream data distributions. We bolded the higher downstream accuracy obtained on the encoders trained with fewer training data points than their full training set. We further systematically decrease the number of samples for the encoder training and until the performance degrades below the one for the training on the full data. With <16k training samples vs. the full 25k, the performance is significantly worse. This result shows that our method can be used to obtain coresets, i.e., smaller subsets of data that yield performance that matches and even outperforms the performance of the encoder trained on the entire dataset.
>
> | **Retained Points ** | **CIFAR10**       | **CIFAR100**   | **STL10**      |
> |----------------------|-------------------|----------------|----------------|
> | 25k (full encoder)   | 63.3% ± 0.92%     | 61.1%±1.14%    | 61.6%±0.83%    |
> | 24k (most memorized) | **64.4% ± 1.03%** | **61.3±0.98%**     | **61.7±1.18%**     |
> | 22k (most memorized) | **63.8% ± 0.76%**     | **61.8±1.24%** | **62.4±1.05%** |
> | 20k (most memorized) | 63.2% ± 1.07%     | 60.8%±0.68%    | 61.1±1.05%     |
> | 16k (most memorized) | 61.8 ± 1.11%      | 58.4%±0.91%    | 59.9±0.89%     |
> | 12k (most memorized) | 59.7% ± 0.74 %    | 55.6%±1.32%    | 55.2±1.24%     |
>
> We added the above results in the new Appendix Section D.5 titled “Practical Implications of Memorization” of the updated version of our paper.

---

> ### Comment · Reviewer_ukN9 · 2023-11-23
> **Acknowledging the responses**
>
> I have checked all the responses and most of my previous concerns are properly addressed. I have raised my score towards acceptance.

---

### Official Review · Reviewer_mj5F · 2023-10-29

**Soundness:** 2 fair
**Presentation:** 3 good
**Contribution:** 2 fair
**Rating:** 6
**Confidence:** 4

**Summary:**

The authors address an interesting issue in self-supervised learning (SSL): the impact of memorization on both SSL and its effects on downstream tasks. They examine the dissimilarity in the representations alignment between augmented views produced by encoders trained on a specific point and those that were not. This problem statement holds relevance in light of the current privacy concerns surrounding deep learning models

**Strengths:**

- It is true that there are not many works addressing the memorization vs. generalization aspect of SSL. This paper aims to address this important issue.

- The proposed method is agnostic to SSL methods and augmentations, making it applicable in various cases.

- It is an insightful study in the right direction. Understanding memorization significantly impacts the generalization of SSL models. The authors have done an excellent job with their extensive empirical analysis.

**Weaknesses:**

- There are several works discussing the issue of memorization in self-supervised learning, which haven't received thorough attention in the literature review. While I understand that not all of these works are directly related, a broader discussion of related literature would enhance the paper's readability (e.g., [1, 2]).

- In the machine learning research community, it's generally accepted that there exists a tension between memorization and generalization. However, I would appreciate more clarity and intuition from the authors regarding why memorization leads to improved generalization.

- Although the memorization score is simple, model- and augmentation-agnostic, it lacks intuitive explanations and theoretical background. It would be beneficial for the authors to provide reasons for why this score measures memorization and offer any applicable theoretical support.

- Regarding the statement, "Our definition compares the difference in alignment of representations for data points and their augmented views returned by both encoders that were trained on these data points and encoders that were not," I'd like to seek clarification. It seems that the training sets of g and f overlap. Could you please provide a justification for this design choice? I had expected a distinct encoder with a different training set or random initialization.

- Considering the proposed metric and previous works in supervised learning, it appears that the contributions are more incremental than novel.

[1] Sadrtdinov, Ildus, Nadezhda Chirkova, and Ekaterina Lobacheva. "On the Memorization Properties of Contrastive Learning." arXiv preprint arXiv:2107.10143 (2021).

[2] Bansal, Yamini, Gal Kaplun, and Boaz Barak. "For self-supervised learning, Rationality implies generalization, provably." ICLR 2020.

**Questions:**

- “we consider a data point as having a high level of memorization by an encoder $f$ if its alignment is significantly higher on $f$ than on encoder g that was not trained with the considered data point” – Why is this the case? Are there any exceptions where this doesn't hold true?

- I understand that the memorization score is relative. However, can one freely choose $g$ as our interest primarily lies in $f$? Have you conducted any ablation studies on the choice of model architectures for $g$ while keeping $f$ constant? Is it necessary for both $g$ and $f$ to share the same architecture?

- What if $g$ is pre-trained on a large but distinct dataset instead of being randomly initialized? Would that result in a reduced memorization score?"

- The training data is divided into 80%, 10%, and 10%. The last two sets do not overlap between $g$ and $f$. How does memorization change when we vary the overlapping ratios from 80% to 70%, 50%, and 30%?

- “We formally verify that data points from Sc (Si ) have statistically significantly higher (lower) memorization scores m than those from Ss and Se.”
“They support the claim that Sc (Si ) is substantially more (less) memorized than Ss and
Se”.
Why is it the case that Sc is substantially more memorized than Ss? Isn't this because memorization is a relative score, and Ss was used to train both $g$ and $f$?

I find this work quite interesting overall. However, in its current form, it lacks sufficient intuition to answer 'why' questions such as:

- Why does memorization lead to generalization?
- Why is the proposed metric suitable for measuring memorization?
- Why were certain design choices made, like training percentages and architecture selections for $g$ and $f$?

The authors need to provide more comprehensive reasoning for their results. Simply presenting empirical findings is insufficient and can lead to confusion for readers.

---

> ### Author Response · Authors · 2023-11-16
> **Thank you for the review, Related Work, Why Memorization leads to Generalization**
>
> We thank the reviewer for the detailed comments and are glad that the reviewer recognizes our work as an “insightful study” of an “interesting issue” and acknowledges our “excellent job with the[ir] extensive empirical analysis”. Below we address all of the points and questions raised by the reviewer one by one.
>
>  >_**There are several works discussing the issue of memorization in self-supervised learning, which haven't received thorough attention in the literature review. While I understand that not all of these works are directly related, a broader discussion of related literature would enhance the paper's readability (e.g., [1, 2]).**_
>
> We would like to thank the reviewer for pointing out these papers from the broader domain of the scope of our work. [1] follows Arpit et al., [2017] in terms of setup and does not formalize a notion of memorization. Instead, they rely on training dynamics to distinguish between examples that are *hard* and *easy to learn* and experimentally evaluate this approach with SimCLR (the SSL framework that offers a form of “classes” through the positive and negative pairs in the mini-batch which is required for the approach from [1]) and the ResNet18 architecture on CIFAR10. In contrast, our work proposes a definition of memorization and empirically evaluates the metric on a broad range of different SSL frameworks, architectures, and datasets.
> [2] studies the impact of pre-training with SSL on the downstream generalization in the form of generalization gaps between training and test data. The factor they identify as most relevant for the generalization gap is “memorization”. Yet, they consider memorization over the entire resulting model (encoder + added downstream classifier), and find that it is bound by the complexity of the downstream classifier. In contrast, we are concerned with memorization of the SSL encoder only and measuring it on the representations, independent of any downstream task. We added discussion on [1,2] as an extension to the related work section of our updated paper to follow the reviewer’s advice.
>
>
> >_**In the machine learning research community, it's generally accepted that there exists a tension between memorization and generalization. However, I would appreciate more clarity and intuition from the authors regarding why memorization leads to improved generalization.**_
>
> We thank the reviewer for this comment and are happy to provide more intuition. As stated by Feldman [2020], most data distributions exhibit a “long tail”, i.e., contain different small sub-populations.
> In Appendix D.4, we show that the memorized samples (from these small sub-populations) of the encoder’s training data lead to a lower alignment loss in larger regions of the representation space. Figure 11 shows that without the inclusion of memorized samples (Figure 11c), the alignment loss is only low in the regions where the main body of the data distribution clusters. However, with the memorized (long-tail, atypical) samples (Figure 11b), larger regions of the representation space exhibit a lower alignment loss. A low alignment loss means that similar inputs obtain similar representations, which helps a downstream classifier to learn better decision boundaries and classify samples in these regions better which boosts generalization.

---

> ### Author Response · Authors · 2023-11-16
> **Intuition, Design Choices, Choice of g**
>
> >_**Although the memorization score is simple, model- and augmentation-agnostic, it lacks intuitive explanations and theoretical background. It would be beneficial for the authors to provide reasons for why this score measures memorization and offer any applicable theoretical support.**_
>
> In Appendix D.3 of our original submission, we provided the intuition behind why it is meaningful to define memorization based on the alignment loss on training data points and to use the leave-one-out style definition.
>
> In summary, the reasoning of our definition then follows Feldman [2020] who, to study the memorization of individual data points in supervised learning, analyzes how much the presence or absence of the data point changes the outcome of the algorithm in terms of prediction loss. This motivates the leave-one-out definition. We maintain the structure of the definition, but instead of considering the prediction loss, rely on the alignment loss.
>
> When we train an encoder f until convergence, we assume that it exhibits a strong alignment on its training data points, i.e., the alignment loss between two different augmented views of any training data point is smaller than a constant c (see Definition 1). A minimized alignment loss in the representation space is a direct effect of the training over all SSL frameworks as shown by prior work. (SSL training minimizes alignment loss over the training data points in the representation space while supervised learning minimizes  the prediction loss on the correct class label.) If we now train two encoders f and g (whose training data differs only in one data point $x$), and consider their difference in alignment loss, we can measure how much impact $x$ has on the training of the algorithm. Large impact on the algorithm and its outcome indicates higher memorization.
>
> >_**Regarding the statement, "Our definition compares the difference in alignment of representations for data points and their augmented views returned by both encoders that were trained on these data points and encoders that were not," I'd like to seek clarification. It seems that the training sets of g and f overlap. Could you please provide a justification for this design choice? I had expected a distinct encoder with a different training set or random initialization.**_
>
> We thank the reviewer for this question and are happy to clarify this design choice: Please keep in mind that we are interested in memorization of individual data points, i.e., in the question, how a given individual data point “impacts” the training. Therefore, the cleanest form to experimentally assess our memorization from Equation 2 would be to have f and g identical in architecture and training dataset with the simple difference being the *one data point $x$* whose memorization we would like to measure. This data point would be included in the dataset of f but not of g.  By then comparing f and g, we would bet the most accurate estimate of the memorization on $x$.
>
> Unfortunately, since SSL models are very large, training many f and many g that each differ in one $x$ only is computationally prohibitive. Therefore, we approximate the memorization by making f and g as close as possible (same architecture, same training), but differing in a few points at the same time. By then comparing f and g’s behavior on all those points, we can make approximate statements on the memorization of these points.
> We discussed all the details of this trade-off and the alternative design choices in Appendix A.1 of our original submission.
>
>
>
> >_**I understand that the memorization score is relative. However, can one freely choose g  as our interest primarily lies in f Have you conducted any ablation studies on the choice of model architectures for g while keeping f constant? Is it necessary for both f and g to share the same architecture?**_
>
> In a similar vein to our previous answer, we would like to note that one cannot freely choose g when interested in measuring memorization of f. As can be seen in our definition of the memorization score (Definition 2), both models f and g are trained with the same learning algorithm A (which encompasses the choice of model architecture). The only intended difference between models f and g is in their training dataset with model g not including datapoint x, which allows to quantify exactly the effect of $x$ on the training algorithm and, thereby, lets one assess the memorization.

---

> ### Author Response · Authors · 2023-11-16
> **Pre-Training of g, High Memorization, Overlapping Data between f and g, Different Data-Subsets**
>
> >_**What if g is pre-trained on a large but distinct dataset instead of being randomly initialized? Would that result in a reduced memorization score?"**_
>
> As mentioned in the previous response, everything for models f and g is kept the same except for the training dataset. Therefore, if model g is pre-trained, then model f must also be pre-trained. Over the dataset of interest for quantifying memorization, this is not likely to have an effect on the types of scores obtained.
>
> >_**“we consider a data point as having a high level of memorization by an encoder f if its alignment is significantly higher on f than on encoder g that was not trained with the considered data point” – Why is this the case? Are there any exceptions where this doesn't hold true?**_
>
> There are no exceptions to the above since the memorization score will only be positive if the alignment loss of model f is lower than the alignment loss on model g. Additionally, the memorization score will be higher the greater the difference of alignment losses between f and g is.
> This can be explained as follows: If we have a data point $x$  which is very common (i.e., similar to other training data points), then the models f and g will both have a good alignment (low alignment loss) on the data point. The alignment loss of f will be low because it was trained on $x$, the alignment loss of g will be low  since g learned on the many similar data points to $x$. In that case, f and g both have a low alignment loss, and we measure a low memorization (difference in alignment losses). However, if the point $x$ is very different from all other points, only when the point is in the training set,  the model can achieve a low alignment loss. Hence, f will have low alignment loss, and g will not, which results in a high difference between the two, and hence a high memorization. Figure 2a of our original submission visualizes this effect.
> We see that only data points that have significantly lower alignment loss in f than in g have a high memorization score.
>
> >_**The training data is divided into 80%, 10%, and 10%. The last two sets do not overlap between f and g. How does memorization change when we vary the overlapping ratios from 80% to 70%, 50%, and 30%?**_
>
> We selected the 80% overlapping ratio to approximate well the precise measure. Ultimately, if we had unlimited compute resources (as we point out in the previous answers), we would train encoders f and g that only differ in a single data point, and repeat that experiment for a significant number of different data points. However, due to the high computational complexity of SSL training, we have to approximate our memorization score (see Appendix A.1). We carried out additional experiments to showcase that the memorization score does not change with higher overlapping ratios (85%) but decreases for smaller ratios (70%). Thus, the ratios below 80% do not provide us with a sufficiently precise measure of memorization. We repeated experiments from Table 1 with ResNet50 trained with SimCLR on CIFAR10 with different splits for the overlap (70% overlap, and 85% overlap). For the best comparability, we made sure to have the same number of training data points over all setups (45k).
>
> | Setup: Overlap (%) for S_S, counts for S_C, and C_I | Avg. Mem. score  | Acc (%)        |
> |-----------------------------------------------------|------------------|----------------|
> | S_S=35k, S_C=10k, S_I=10k                           | 0.325 ± 0.008    | 77.95% ± 1.23% |
> | S_S=40k, S_C=5k, S_I=5k (See Table 1)               | 0.339 ± 0.011    | 77.12% ± 1.42% |
> | S_S=42.5k, S_C=2.5k, S_I=2.5k                       | 0.337 ± 0.010    | 76.84% ± 0.85% |
>
>
> >_**“We formally verify that data points from Sc (Si ) have statistically significantly higher (lower) memorization scores m than those from Ss and Se.” “They support the claim that Sc (Si ) is substantially more (less) memorized than Ss and Se”. Why is it the case that Sc is substantially more memorized than Ss? Isn't this because memorization is a relative score, and Ss was used to train both f and g ?**_
>
> Indeed, the reviewer is perfectly correct with this conclusion. $S_C$ is used only as training data for f, hence f will have a low alignment loss (while g will not necessarily) which results in higher memorization. $S_S$ is training data of both encoders, hence both will have a relatively low alignment loss for both models, and hence, the difference between the alignment losses, i.e., the memorization, will be low.

---

> ### Author Response · Authors · 2023-11-16
> **General Comments and Summary**
>
> >_**I find this work quite interesting overall. However, in its current form, it lacks sufficient intuition to answer 'why' questions such as: Why does memorization lead to generalization? Why is the proposed metric suitable for measuring memorization? Why were certain design choices made, like training percentages and architecture selections for g and f?**_
>
> We thank the reviewer for this comment and summarize our answers below. For more details, please see our answers to the reviewer’s comments and questions above:
> - The intuition behind why memorization leads to better generalization can be found in Appendix D.3 and D.4. Memorized (atypical) samples lower the alignment loss over larger regions of the representation space. This leads to more consistent representations in these regions which facilitates learning of the downstream classifiers.
> - Our metric is suitable to measure memorization because it captures the impact that a single data point makes on the trained encoder. Our work is based on the prior research on memorization in supervised learning by Feldman [2020] and Feldman and Zhang [2020]. While in supervised learning, the impact can be measured w.r.t. to the model’s prediction on the data point’s ground truth label, such labels do not exist in SSL. Instead of optimizing for labels, all SSL frameworks implicitly or explicitly optimize for alignment, which makes alignment difference the suitable metric to assess the impact on the trained encoder.
> - The design choices made in the paper are discussed in detail in Appendix A.1. Taking the trade-offs between an *accurate* and *computationally possible* approximation into account, we chose g as close as possible to f (same architecture) with as much data overlap as possible. Our additional experiments presented in this answer highlight that our choice of percentage yields very close results to larger overlap in percentage (with desired but computationally infeasible overlap of near 100%, difference in just one data point). Hence, our approximation is sufficiently accurate.

---

> ### Comment · Reviewer_mj5F · 2023-11-22
> **Response to authors' rebuttal**
>
> I appreciate the authors' efforts in addressing most of the concerns I raised earlier. However, I still find some gaps in the reasoning behind their results. Nevertheless, the extensive experimental analysis presented in the paper is noteworthy. Despite these concerns, I recognize the potential value of this work as a useful resource for readers. Therefore, I have adjusted my score to 6 and support the paper's acceptance.

---

> > ### Author Response · Authors · 2023-11-22
> > **Thank you for the review**
> >
> > We would like to thank the reviewer for their positive feedback and their engagement in our rebuttal. If there are any specific gaps that we could further elaborate on, we are very happy to do so.

---

### Official Review · Reviewer_tPGb · 2023-10-30

**Soundness:** 3 good
**Presentation:** 3 good
**Contribution:** 3 good
**Rating:** 8
**Confidence:** 4

**Summary:**

The paper proposes a way to measure memorization at the representation level, which is applicable to SSL approaches, in contrast to previous work quantifying memorization in supervised learning. The memorization metric is based on measuring differences in alignment between different views of the same input point, between models trained with and without the specific point.
With their new measure, the authors investigate the degree of memorization in encoder models using different architectures, trained on different datasets and using different SSL approaches. They find that SSL-trained models exhibit memorization and that the degree of memorization benefits downstream performance.

**Strengths:**

- Relevance: Understanding memorization is an important problem, which is challenging and underexplored in the SSL domain. The paper makes an important contribution in conceptualizing memorization in this space, as well as proposing a corresponding measure. Further, a representation-level measure of memorization is a valuable tool that could be applied in other interesting ways as well, such as localizing memorization in models.
- Soundness: The experiments are thorough and the methodology is solid.
- Presentation: The paper is well written and easy to follow.

### Neutral:
- Novelty: The findings seem to be similar to those made for supervised learning, i.e. that memorization benefits generalization, as well as atypical points exhibiting more memorization. However, SSL approaches use a different learning paradigm, so it is not a priori clear whether one should expect similar trends to hold. It is therefore interesting to see that similar dynamics hold for SSL models as well.

**Weaknesses:**

- The results seem to show that memorization benefits downstream performance. However, *why* this is happening is not quite clear to me. For instance, when removing points with high memorization scores from the training data, does the performance of the model primarily drop on points "similar" to the removed ones? If this was the case, memorization here might actually be more of a long-tail generalization phenomenon.
- In Section 4.4, Eq. (3) you limit the degree of alignment between representations. However, in addition to reducing memorization, this intervention might also degrade the representations in other ways. Therefore, it's not clear to me whether we can conclude that a reduction in memorization is causing a drop in accuracy, or whether both might just be consequences of a degradation in representation quality due to the regularization.
- There are some smaller clarity issues:
    1. In Section 4, first paragraph, what is the normalization procedure applied to constrain memorization scores between -1 and 1?
    2. You say that experiments are repeated over three independent seeds. Does that mean you use different data splits for each seed or just different weight initializations of the models?
    3. Table 1, what is "Frac. Mem."? Is it the same as "Avg. Mem." defined in the caption?
    4. In 4.4, do you remove the 500, 1K, etc. datapoints with highest memorization from the set of 25K points, or from the full CIFAR10 training set?
    5. In 4.4, why do you use cosine similarity here vs l2 distance earlier?
    6. In 4.5, what does the term "exploited" in the context of Deja Vu mean? Would you expect MAE to exhibit higher memorization?

**Questions:**

- Does the metric agree with previous metrics for quantifying memorization in supervised learning? I.e. given a supervised learning model (where you can apply prior supervised learning work), would the metric highlight the same points as memorized as previous metrics for supervised learning?
- How dependent is the metric on the types of data augmentations used? I.e. if a model was trained with masking, would the metric also be able to quantify memorization under e.g. rotation or noise augmentations?
- What are atypical datapoints? Is the judgement of typicality just based on visual inspection or are there other indicators as well?
- The idea behind the metric is that quantifying memorization via alignment differences between different augmentations of the same input point is the common denominator between different SSL approaches (contrastive, non-contrastive, reconstruction-based). Would it be possible to define a "stronger" notion of memorization if one were to only consider one family of SSL approaches, e.g. contrastive ones?

### Suggestions:
- Giving the memorization metric a name would make it easier to refer to it.
- Given that the proposed metric operates at the representation level, it might be interesting to quantify memorization at intermediate layers in the model, to potentially localize where it is happening.

---

> ### Author Response · Authors · 2023-11-14
> **Thank you for the Review, Performance, Normalization**
>
> We appreciate the positive feedback and thank the reviewer for their detailed comments on our work. We are glad that the reviewer recognizes our work as an “important contribution in conceptualizing memorization” in SSL and as “a valuable tool”. We hope that our work will contribute to providing an understanding of memorization in the SSL domain. Below, we offer clarifications on the points raised by the reviewer on a question-by-question basis.
>
> >_**When removing points with high memorization scores from the training data, does the performance of the model primarily drop on points "similar" to the removed ones?**_
>
> We would like to highlight that, in contrast to most prior work, we do not focus our study on the limited setup where the training data of the SSL encoder is the same as the training data of the downstream classifier (see for example Figure 3, where we remove the most memorized data points of an encoder trained on CIFAR10, and observe a drop in downstream accuracy for CIFAR100 and STL10 classifiers). The expressiveness of “similarity” for assessing performance is not straightforward in this setup with data from different distributions and with different classes and their number.
>
> However, what we show in Appendix D.3 and formalize in Appendix D.4 is that the atypical memorized samples of the encoder’s training data lead to a lower alignment loss in larger regions of the representation space. Figure 10 shows that without the inclusion of memorized samples (Figure 10c), the alignment loss is only low in the regions where the main data distribution clusters. However, with the memorized samples (Figure 10b), larger regions of the representation space exhibit a lower alignment loss. A low alignment loss means that similar inputs obtain similar representations, which helps a downstream classifier to learn better decision boundaries and classify samples in these regions better which boosts generalization.
>
>
> >_**In Section 4, first paragraph, what is the normalization procedure applied to constrain memorization scores between -1 and 1**_
>
> We added an explanation of the normalization procedure in Appendix A of the updated paper. We normalize (using the $\ell_2$ norm) the representations output by encoders $f$ and $g$. Then, we calculate the differences in alignment loss per data sample over both encoders. We then normalize these differences by dividing them by the range (largest minus smallest difference), and report the memorization score as the average of the resulting scores over all data points in $S_C$.

---

> ### Author Response · Authors · 2023-11-14
> **Independent Seeds, Caption, Data Point Removal, Cosine Similarity, Deja Vu, Previous Metrics**
>
> >_**You say that experiments are repeated over three independent seeds. Does that mean you use different data splits for each seed or just different weight initializations of the models?**_
>
> We preserve the same data splits and repeat the experiments over three independent seeds with different random model initializations. By keeping the data splits constant, we enable comparability between the different approaches and experiments since in different data splits, we might encounter more or less atypical data, which might result in highly different results for memorization.
>
> >_**Table 1, what is "Frac. Mem."? Is it the same as "Avg. Mem." defined in the caption?**_
>
> We would like to thank the reviewer for pointing out this inconsistency. Indeed, this is a typo and should be Avg. Mem. Following the reviewer’s suggestion, we gave a name to our metric (SSLMem), and now refer to our results as that in the updated version of the paper.
>
> >_**In 4.4, do you remove the 500, 1K, etc. datapoints with highest memorization from the set of 25K points, or from the full CIFAR10 training set?**_
>
> We removed them from the 25k data points. This is because only these 25k were used to train the SSL encoder and we measured the memorization of the encoder on these training data points.
>
> >_**In 4.4, why do you use cosine similarity here vs l2 distance earlier?**_
>
> We used these two metrics interchangeably since the representations are normalized and L2-Distance(u,v) = 2 * (1- Cosine-Similarity(u,v)). To make it consistent, we will change the Cosine-Similairty to L2-Distance in the experiment.
>
> >_**In 4.5, what does the term "exploited" in the context of Deja Vu mean? Would you expect MAE to exhibit higher memorization?**_
>
> The Deja Vu memorization is designed to rely on crops, namely it measures the relation between a crop of the background with the entire image. Cropping is a core feature of MAE training. Hence, we expected the Deja Vu memorization metric to yield high scores for MAE. This does not necessarily mean that MAE exhibits the highest memorization but it means that the Deja Vu metric can capture that behavior of MAE.
>
> >_**Does the metric agree with previous metrics for quantifying memorization in supervised learning? I.e. given a supervised learning model (where you can apply prior supervised learning work), would the metric highlight the same points as memorized as previous metrics for supervised learning?**_
>
>
> To provide an answer to the reviewer’s insightful question, we implemented an approximation of the leave-one-out memorization score by Feldman [2020]. Therefore, we trained a model $f$ and a model $g$ (both ResNet50 on CIFAR10) in a supervised manner. For best comparability with our results, we chose 40k data points overlap between the two models and 5k difference. On the 5k data points used to train f but not g, we calculated the difference in softmax outputs between f and g. The data points with the highest difference are the ones with the highest memorization according to Feldman [2020]. To calculate our metric on the same models, we removed the classification layer and then calculated our metric on the output representations. In the updated paper, we show in Figure 8 of our updated paper, the first 10 most memorized images from the supervised and the self-supervised models. We observe a substantial overlap.
> These preliminary results indicate a similarity between the two metrics. We hypothesize, based on our experiments, that both metrics (from Feldman [2020] and our) report similar atypical samples as memorized.

---

> ### Author Response · Authors · 2023-11-14
> **Augmentations, Atypical Data Points, Notion of Memorization, Metric Name, Localizing Memorization**
>
> >_**How dependent is the metric on the types of data augmentations used? I.e. if a model was trained with masking, would the metric also be able to quantify memorization under e.g. rotation or noise augmentations?**_
>
> The metric is not dependent on the types of data augmentations but we use exactly the same augmentation set for training and measurement of memorization. This is because the training optimizes model behavior while applying specific augmentations and we are interested in precisely capturing the impact that training leaves on the SSL model.
>
> Minor note: it was also pointed out as a strength by Reviewer mj5F that “The proposed method is agnostic to SSL methods and augmentations, making it applicable in various cases.”
>
> We also ran additional experiments to show that using different augmentations for measurement of memorization and for training lowers the memorization score. We experimented with ResNet50 trained on CIFAR10 by SimCLR (77.12% accuracy on the downstream classifier). SimCLR originally uses the following augmentations: RandomResizedCrop(32), RandomHorizontalFlip(p=0.5), ColorJitter(0.4, 0.4, 0.4, 0.1)], p=0.8), and RandomGrayscale(p=0.2).
> We used those for training but measured memorization using different augmentations:
>
> | Augmentation for Measurement                     | Average SSLMem Memorization |
> |-----------------------------------|-----------------------------|
> | SimCLR original                   | 0.339 ± 0.011               |
> | GaussianNoise (mean=0 and std=0.2) | 0.321 ± 0.014               |
> | Rotate 90°                        | 0.308 ± 0.009               |
> | Rotate 270°                       | 0.328 ± 0.011               |
> | ColorDrop 0.25                   | 0.298 ± 0.006               |
>
> The results indicate that using the same original augmentations that were used during training also for measuring memorization yields the highest memorization score, i.e., gives the strongest signal to measure memorization. Yet, the other augmentations’ scores are not significantly different, and hence can be used equally to approximate the degree of memorization.
>
> >_**What are atypical datapoints? Is the judgement of typicality just based on visual inspection or are there other indicators as well?**_
>
> We refer to atypical data points as data points that are uncommon in the data distribution and different in terms of their features. Visual inspection is indeed the most straightforward way to judge how atypical a data point is. We present Figure 1 to allow for such a visual inspection.  However, other factors, such as distance in the image space or the similarity between extracted features of the data point may also be used. Those could be assessed using, for example, clustering approaches. We highlighted in the updated version of the paper that we refer to our visual inspection in that context.
>
>
>  >_**The idea behind the metric is that quantifying memorization via alignment differences between different augmentations of the same input point is the common denominator between different SSL approaches (contrastive, non-contrastive, reconstruction-based). Would it be possible to define a "stronger" notion of memorization if one were to only consider one family of SSL approaches, e.g. contrastive ones?**_
>
> Indeed, it might be possible to define a stronger notion of memorization if the types of SSL approaches are restricted. For example, in the case of contrastive approaches, one may incorporate the uniformity metric as both alignment and uniformity are optimized in this SSL type. However, the main issue is that memorization should be defined per data point while uniformity depends on all the training data points at the same time.
>
> >_**Giving the memorization metric a name would make it easier to refer to it.**_
>
> We thank the reviewer for this valuable suggestion. We modified the submission accordingly and referred to our metric as “SSLMem”.
>
> >_**Given that the proposed metric operates at the representation level, it might be interesting to quantify memorization at intermediate layers in the model, to potentially localize where it is happening.**_
>
> This is a very valuable observation and we will make sure to follow up on this work to pinpoint where the memorization happens in the SSL models.

---

> > ### Author Response · Authors · 2023-11-18
> > **Additional Experimental Results on the Localization of Memorization**
> >
> > >_**Given that the proposed metric operates at the representation level, it might be interesting to quantify memorization at intermediate layers in the model, to potentially localize where it is happening.**_
> >
> > We also ran additional experiments to assess this question. We measured memorization on the ResNet50, trained with SimCLR on CIFAR10. From the initial experiments, we can observe that the memorization happens in the last layers of the network.
> >
> > | **Layer Block (ReNet50 has 4 blocks)** | **Memorization Score** |
> > |:--------------------------------------:|:----------------------:|
> > |                    1                   |      0.048 ± 0.009     |
> > |                    2                   |      0.101 ± 0.012     |
> > |                    3                   |      0.166 ± 0.009     |
> > |                    4                   |      0.275 ± 0.016     |
> > |         Output representations         |     0.339 ± 0.011      |

---

> ### Comment · Reviewer_tPGb · 2023-11-21
>
> Thanks a lot for the extensive response and all the additional results and clarifications! They address most of the issues and questions I had about the work. I will maintain my already positive score but raise my confidence level.
>
> It is good to see that SSLMem seems to agree with existing measures of memorization. This raises confidence in the measure. My suggestion would be to complement the qualitative analysis in Figure 8 with a quantitative check, e.g. by computing the rank correlation between SSLMem and Feldman (2022). That would provide a better understanding of how well the trend holds beyond the high end of the spectrum.
> It is also interesting to see that memorization seems to be increasing towards the output layer, and that memorization under one set of augmentations seems to somewhat extend to other augmentations.

---

> > ### Author Response · Authors · 2023-11-22
> > **Thank you for your positive feedback and the engagement in the rebuttal**
> >
> > We are glad that our rebuttal addressed most of the reviewer's questions. We appreciate that the reviewer maintained the high score and increased the confidence level. We will further follow the reviewer's suggestion and will compute the rank correlation between SSLMem and Feldman (2020).

---

> > > ### Author Response · Authors · 2023-11-22
> > > **Rank Correlation between SSLMem and Feldman (2020)**
> > >
> > > As promised to the reviewer, we calculated the rank correlation between SSLMem and Feldman (2020). Additionally, as a more intuitive metric, we also calculated the percentage of overlap in most memorized samples over different numbers of most-memorized samples by both metrics.
> > >
> > > We present the results in the updated version of the paper in Appendix B.3, Table 12. The results indicate that there is a roughly 50% overlap between the most-memorized samples identified by both methods, and that the ranking between the samples is similarly consistent as the rankings by our SSLMem method over different SSL frameworks (see Table 13).

---

### Official Review · Reviewer_fWKE · 2023-11-22

**Soundness:** 3 good
**Presentation:** 3 good
**Contribution:** 3 good
**Rating:** 6
**Confidence:** 3

**Summary:**

In this paper the authors propose SSLMem, a framework for defining memorization within Self Supervise Learning (SSL). The authors base their framework analyzing the  difference in alignment of representations for data points and their augmented views returned by encoders. They show an empirical analysis on diverse encoder architectures and datasets, highlighting that significant fractions of training data points experience memorization, and highlight that memorization is essential for encoders to achieve generalization performance on downstream tasks.

**Strengths:**

-good presentation and writhing

-easy flow of argumentation

-interesting and valuable insights related to the interplay between memorization and generalization

**Weaknesses:**

-importance and influence of the augmentations, it would have been nice to see does a particular or a set of augmentations plays a role in this empirical evaluation, also instead of the augmentations, how similar data samples play a role it would be interesting to analyze

-regarding the experiment considering differential privacy, only one algorithm was evaluated, I was not able to see other evidence (evaluation using different setups and algorithms) that supports the case about memorization in this context

**Questions:**

Does a particular augmentation or a set of augmentations plays a role in this empirical evaluation?

How does similar data samples play a role with respect to the memorization vs generalization claims?

---

> ### Author Response · Authors · 2023-11-22
> **Answer to Reviewer fWKE: Augmentations, Similar Data Samples, and Differential Privacy**
>
> We would like to thank the reviewer for their review and positive feedback on our submission. In the following, we address the points raised one-by-one.
>
> >_**importance and influence of the augmentations, it would have been nice to see: does a particular or a set of augmentations plays a role in this empirical evaluation**_
>
> With respect to the role of augmentations, we performed additional experiments during the rebuttal period on assessing memorization with different sets of augmentations than the ones used during training. The results can be found in the updated Appendix B.1, especially Table 8. The results indicate that using the same original augmentations that were used during training also for measuring memorization yields the highest memorization score, i.e., gives the strongest signal to measure memorization. Yet, the other augmentations’ scores are not significantly different and hence can be used equally to approximate the degree of memorization.
>
> >_**also instead of the augmentations, how similar data samples play a role it would be interesting to analyze**_
>
> We would like to highlight that, in contrast to most prior work, we do not focus our study on the limited setup where the training data of the SSL encoder is the same as the training data of the downstream classifier (see, for example, Figure 3, where we remove the most memorized data points of an encoder trained on CIFAR10, and observe a drop in downstream accuracy for CIFAR100 and STL10 classifiers). The expressiveness of “similarity” for assessing performance is not straightforward in this setup with data from different distributions, with different classes and their number.
>
> However, what we show in Appendix D.3 and formalize in Appendix D.4 is that the atypical memorized samples of the encoder’s training data lead to a lower alignment loss in larger regions of the representation space. Figure 10 shows that without the inclusion of memorized samples (Figure 10c), the alignment loss is only low in the regions where the main data distribution clusters. However, with the memorized samples (Figure 10b), larger regions of the representation space exhibit a lower alignment loss. A low alignment loss means that similar inputs obtain similar representations, which helps a downstream classifier to learn better decision boundaries and classify samples in these regions better, which boosts generalization.
>
> >_**regarding the experiment considering differential privacy, only one algorithm was evaluated, I was not able to see other evidence (evaluation using different setups and algorithms) that supports the case about memorization in this context**_
>
> We would like to point out that the differential private stochastic gradient descent (DP-SGD) is the de facto standard algorithm for performing machine learning models with privacy protection. Therefore, we evaluated with this algorithm. Differential privacy (implemented through DP-SGD), by definition, limits the degree of memorization since it limits the impact that any individual data point is allowed to have on the model updates during training. We evaluated different privacy setups ($\varepsilon=\infty$, i.e., no privacy protection, $\varepsilon=20$, i.e. weak privacy protection, and $\varepsilon=8$, i.e., stronger privacy protection).

---

### Author Response · Authors · 2023-11-18
**Highlights of the Rebuttal**

We would like to thank all reviewers for their insightful comments and suggestions. We addressed all of them one-by-one in the individual answers to the reviewers. In the following, we summarize the highlights of the updated version of our paper.

1. **Core sets:**  Based on the suggestion by Reviewer ukN9, we ran additional experiments that highlight the ability of our method to identify core sets. The results in our new Appendix D.5 show that when training only on the most memorized samples identified by our method, we can outperform the encoder trained on the entire dataset.
2. **New empirical results:** Based on the suggestion by Reviewer ukN9, we ran additional experiments on more and complex datasets (ImageNet, Food-101, and Flower102) and more architectures (ResNet34, Wide-ResNet). All results can be found in Appendix B.5 of the updated paper. In summary, our new results show that on the more complex datasets, the performance gap when removing memorized vs. random data points is even more significant than on the coarse-grained datasets from our original Figure 3. Additionally, we observe that the SSL memorization increases with the larger number of parameters (for both width and depth of an encoder).
3. **More downstream tasks:** In addition to classification and segmentation downstream tasks from the original submission, we ran new experiments for *depth estimation*. Our results in Appendix B.2 show that the conclusion that memorization improves downstream generalization also holds for this new task.
4. **Comparing memorization scores for SSL vs supervised learning:** Thanks to the suggestion by Reviewer tPGb, we assessed the consistency between our metric and the metric from supervised learning (by Feldman [2020] and Feldman and Zhang [2020]). Our new results in Appendix B.3 suggest that both metrics report similar atypical samples as memorized.
5. **Ablation on our design choice:** Thanks to the suggestion by Reviewer mj5F, we added additional ablations in Appendix A.1 that highlight that our empirical way to approximate memorization is valid, and that in particular the design choices on the overlap of training data between encoder $f$ and $g$ is large enough to yield a high fidelity approximation while being computationally efficient by supporting the assessment of multiple data points’ memorization at once.
6. **Localization of memorization:** Also based on the  suggestion by Reviewer tPGb, we performed additional experiments to localize memorization. Our new insights suggest that memorization occurs in the last layers of encoders.

Finally, we are happy to engage in a discussion with the reviewers and answer additional questions.

---

### Meta-Review · Area_Chair_FXT3 · 2023-12-09

**Metareview:**

The authors investigate the impact of memorization on self-supervised learning (SSL) and its repercussions on downstream tasks. They scrutinize the dissimilarity in representation alignment between augmented views generated by encoders trained on specific points and those that were not. This exploration gains significance in the context of contemporary privacy concerns surrounding deep learning models. In some sense, the paper introduces a novel framework defining memorization in SSL, comparing representation alignment for data points and their augmented views between encoders trained on these points and those that were not. Empirical analyses across diverse encoder architectures and datasets reveal substantial instances of memorization in SSL, highlighting its essential role for achieving higher generalization performance on various downstream tasks.

The reviewers are unanimously positive about the paper’s technical merits and solidness.

**Justification For Why Not Higher Score:**

The paper’s impact is limited by a deeper understanding for why memorization induced from SSL benefits downstream performances. The findings for the memorization effects of SSL are surprisingly similar to those made of supervised learning - it is a bit unclear what to take away from this contribution.

**Justification For Why Not Lower Score:**

The presented results are solid and do offer useful insights for the memorizations induced by SSL. The reviewers are unanimously positive about the paper's contribution.

---

### Decision · Program_Chairs · 2024-01-16

Accept (poster)